# Finding the Global Semantic Representation in GAN through Fréchet Mean

**Jaewoong Choi**
Korea Institute for Advanced Study
chjw1475@kias.re.kr

**Geonho Hwang, Hyunsoo Cho, Myungjoo Kang**[*]
Seoul National University
{hgh2134,hscho100,mkang}@snu.ac.kr

## Abstract

The ideally disentangled latent space in GAN involves the global representation of latent space with semantic attribute coordinates. In other words, considering that this disentangled latent space is a vector space, there exists the global semantic basis where each basis component describes one attribute of generated images. In this paper, we propose an unsupervised method for finding this global semantic basis in the intermediate latent space in GANs. This semantic basis represents sample-independent meaningful perturbations that change the same semantic attribute of an image on the entire latent space. The proposed global basis, called Fréchet basis, is derived by introducing Fréchet mean to the local semantic perturbations in a latent space. Fréchet basis is discovered in two stages. First, the global semantic subspace is discovered by the Fréchet mean in the Grassmannian manifold of the local semantic subspaces. Second, Fréchet basis is found by optimizing a basis of the semantic subspace via the Fréchet mean in the Special Orthogonal Group. Experimental results demonstrate that Fréchet basis provides better semantic factorization and robustness compared to the previous methods. Moreover, we suggest the basis refinement scheme for the previous methods. The quantitative experiments show that the refined basis achieves better semantic factorization while constrained on the same semantic subspace given by the previous method.

## 1 Introduction

Generative Adversarial Networks (GANs, (Goodfellow et al., 2014)) have achieved impressive success in high-fidelity image synthesis, such as ProGAN (Karras et al., 2018), BigGAN (Brock et al., 2018), and StyleGANs (Karras et al., 2019; 2020a;b; 2021). Interestingly, even when a GAN model is trained without any information about the semantics of data, its latent space often represents the semantic property of data (Radford et al., 2016; Karras et al., 2019). To understand how GAN models represent the semantics, several studies investigated the disentanglement (Bengio et al., 2013) property of latent space in GANs (Goetschalckx et al., 2019; Jahanian et al., 2019; Plumerault et al., 2020; Shen et al., 2020). Here, a latent space in GAN is called *disentangled* if there exists an optimal basis of the latent space where each basis coefficient corresponds to one disentangled semantics (generative factor).

One approach to studying the disentanglement property is to find meaningful latent perturbations that induce the disentangled semantic variation on generated images (Ramesh et al., 2018; Härkönen et al., 2020; Shen & Zhou, 2021; Choi et al., 2022b). This approach can be interpreted as investigating how the semantics are represented around each latent variable. We classify the previous works on meaningful latent perturbations into *local* and *global* methods depending on whether the proposed perturbation is sample-dependent or sample-ignorant. In this work, we focus on the global methods (Härkönen et al., 2020; Shen & Zhou, 2021). If the latent space is ideally disentangled, the optimal semantic basis becomes the global semantic perturbation that represents a change in the same generative factor on the entire latent space. In this regard, these global methods are attempts to find the best-possible semantic basis on the target latent space. Throughout this work, the semantic subspace represents the subspace generated by the corresponding semantic basis.

---

[*]Corresponding author

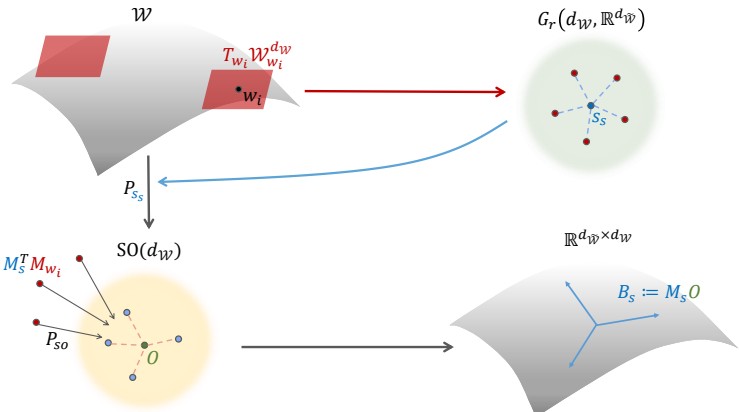

Figure 1: **Overview of Fréchet basis.** The global semantic subspace $\mathcal{S}_s$ is defined as the Fréchet mean of intrinsic tangent spaces $T_{\mathbf{w}_i}\mathcal{W}_{\mathbf{w}_i}^{d_\mathcal{W}}$ in the Grassmannian manifold $Gr(d_\mathcal{W}, \mathbb{R}^{d_{\bar{w}}})$. Fréchet basis $\mathcal{B}_s$ is discovered by selecting the optimal basis of $\mathcal{S}_s$ using the Fréchet mean in the Special Orthogonal Group.

In this paper, we propose an unsupervised method for finding the global semantic perturbations in a latent space in GAN, called **Fréchet Basis**. Fréchet Basis is based on the *Fréchet mean* on the Riemannian manifold. Fréchet mean is a generalization of centroid to the general metric space (Fréchet, 1948) and the Riemannian manifold is the metric space (Lee, 2013). In particular, Fréchet Basis is discovered in two steps (Fig 1). First, we find the global semantic subspace $\mathcal{S}_s$ of latent space as Fréchet mean in the Grassmannian manifold (Boothby, 1986) of the intrinsic tangent spaces. Here, the intrinsic tangent space represent the local semantic subspace (Choi et al., 2022b). Second, Fréchet basis $\mathcal{B}_s$ is discovered by selecting the optimal basis of $\mathcal{S}_s$ via Fréchet mean in the Special Orthogonal Group (Lang, 2012). Our experiments show that Fréchet basis provides better semantic factorization and robustness compared to the previous unsupervised global methods. Moreover, the second step in finding Fréchet basis provides the basis refinement scheme for the previous global methods. In our experiments, the basis refinement achieves better semantic factorization than the previous methods while keeping the same semantic subspace. Our contributions are as follows:

1. We propose unsupervised global semantic perturbations, called Fréchet basis. Fréchet basis is discovered by introducing Fréchet mean to the local semantic perturbations.
2. We show that Fréchet basis achieves better semantic factorization and robustness compared to the previous global approaches.
3. We propose the basis refinement scheme, which optimizes the semantic basis on the given semantic subspace. We can refine the previous global approaches by applying the basis refinement on their semantic subspaces.

## 2 RELATED WORKS AND BACKGROUND

**Latent Perturbation for Image Manipulation** The latent space of GANs often represents the semantics of data even when the model is trained without the supervision of the semantic attributes. Early approaches to understanding the semantic property of latent space showed that the vector arithmetic on latent space leads to the semantic arithmetic on the image space (Radford et al., 2016; Upchurch et al., 2017). In this regard, a line of research has been conducted to find meaningful latent perturbations that perform image manipulation in disentangled semantics. We categorize the previous works into *local* and *global* methods according to sample dependency. The local method finds meaningful perturbations for each latent variable for one semantic attribute, e.g., Ramesh et al. (2018), Latent Mapper in StyleCLIP (Patashnik et al., 2021), Attribute-conditioned normalizing flow in StyleFlow (Abdal et al., 2021), Local image editing in Zhu et al. (2021), and Local Basis (Choi et al., 2022b). By contrast, the global methods offer the sample-independent meaningful perturbations for each latent space, e.g., Global directions in StyleCLIP (Patashnik et al., 2021), GANSpace (Härkönen et al., 2020), and SeFa (Shen & Zhou, 2021). Among them, StyleCLIP is a supervised method requiring text descriptions of each generated image to train CLIP model (Radford et al., 2021). Throughout this paper, we investigate unsupervised methods such as GANSpace and

SeFa. GANSpace (Härkönen et al., 2020) suggested the principal components of latent space obtained by performing PCA as global meaningful perturbations. SeFa (Shen & Zhou, 2021) proposed the singular vectors of the first weight matrix as global disentangled perturbations.

**Unsupervised global disentanglement score**   The disentanglement of latent space is expressed as the correspondence between the semantic attributes of data and the axes of latent space. Because the definition of disentanglement depends on attributes, most of the existing disentanglement metrics for latent spaces are supervised ones, e.g., DCI score (Eastwood & Williams, 2018), $\beta$-VAE metric (Higgins et al., 2017), and FactorVAE metric (Kim & Mnih, 2018). They require the attribute annotations of generated images. This requirement restricts the broad applicability of disentanglement evaluation on real datasets. To address this restriction, Choi et al. (2022a) proposed an unsupervised global disentanglement score, called *Distortion*. Distortion metric measures the variation of tangent space on the *learned latent manifold $\mathcal{W}$*. Hence, Distortion metric relies purely on the geometric property of the latent space and does not require attribute labels.

**Background**   Choi et al. (2022b) suggested a framework for analyzing the semantic property of intermediate latent space by its local geometry. This analysis is performed on the *learned latent manifold $\mathcal{W} = f(\mathcal{Z})$*, where $f$ denotes the subnetwork $f$ from the input noise space $\mathcal{Z}$ to the target latent space. Here, we assume $\mathcal{Z}$ is the entire Euclidean space, i.e., $\mathcal{Z} = \mathbb{R}^{d_{\mathcal{Z}}}$ for some $d_{\mathcal{Z}}$. Note that this is satisfied for the usual Gaussian prior $p(\mathbf{z}) = \mathcal{N}(0, I_{d_{\mathcal{Z}}})$. Choi et al. (2022b) proposed a method for finding the $k$-dimensional local approximation $\mathcal{W}_{\mathbf{w}}^{k}$ of $\mathcal{W}$ around $\mathbf{w} = f(\mathbf{z}) \in \mathcal{W}$. This local approximation $\mathcal{W}_{\mathbf{w}}^{k}$ is discovered by the low-rank approximation problem of $df_{\mathbf{z}}$ and this solution is given by SVD. Then, $\mathcal{W}_{\mathbf{w}}^{k}$ is given as follows: For the $i$-th singular vector $\mathbf{u}_i^{\mathbf{z}} \in \mathbb{R}^{d_{\mathcal{Z}}}$, $\mathbf{v}_i^{\mathbf{w}} \in \mathbb{R}^{d_{\mathcal{W}}}$, and $i$-th singular value $\sigma_i^{\mathbf{z}} \in \mathbb{R}$ of $df_{\mathbf{z}}$ with $\sigma_1^{\mathbf{z}} \geq \cdots \geq \sigma_m^{\mathbf{z}}$ and $m = \min(d_{\mathcal{Z}}, d_{\mathcal{W}})$,

$$df_{\mathbf{z}}(\mathbf{u}_i^{\mathbf{z}}) = \sigma_i^{\mathbf{z}} \cdot \mathbf{v}_i^{\mathbf{w}} \quad \text{for } \forall i, \qquad \text{Local Basis}(\mathbf{w} = f(\mathbf{z})) = \{\mathbf{v}_i^{\mathbf{w}}\}_{1 \leq i \leq n}, \tag{1}$$

$$\mathcal{W}_{\mathbf{w}}^{k} = \left\{ f\left(\mathbf{z} + \sum_i t_i \cdot \mathbf{u}_i^{\mathbf{z}}\right) \mid t_i \in (-\epsilon_i, \epsilon_i), \text{ for } 1 \leq i \leq k \right\}, \tag{2}$$

$$T_{\mathbf{w}} \mathcal{W}_{\mathbf{w}}^{k} = \text{span}\{\mathbf{v}_i^{\mathbf{w}} : 1 \leq i \leq k\}. \tag{3}$$

Choi et al. (2022b) showed that the codomain singular vectors, called Local Basis (Eq 1), serve as the local semantic perturbations around $\mathbf{w}$. In this respect, *the tangent space at $\mathbf{w}$ represents the local semantic subspace because* it is spanned the local semantic perturbations (Eq 3).

Upon this framework, Choi et al. (2022a) proposed the intrinsic local dimension estimation scheme for the latent manifold $\mathcal{W}$ through the robust rank estimate (Kritchman & Nadler, 2008) of $df_{\mathbf{z}}$. Geometrically, the intrinsic local dimension represents the number of dimensions required to properly approximate the denoised $\mathcal{W}$. *Choi et al. (2022a) showed that this local dimension corresponds to the number of local semantic perturbations.* Using this correspondence, Choi et al. (2022a) introduced the unsupervised disentanglement score called *Distortion*. Distortion metric is defined as the normalized variation of intrinsic tangent space on the latent manifold. The normalized variation is expressed as the ratio of the distance between two tangent spaces at two *random* $\mathbf{w} \in \mathcal{W}$ (Eq 4) to the distance between two tangent spaces at two *close* $\mathbf{w}$ (Eq 5). The distance between tangent spaces is measured by the dimension-normalized Geodesic Metric $d_{geo}^{k}$ Choi et al. (2022a) in Grassmannian manifold Boothby (1986). Specifically, ***Distortion*** of $\mathcal{W}$ is defined as $\mathcal{D}_{\mathcal{W}} = I_{rand}/I_{local}$ with

$$I_{rand} = \mathbb{E}_{\mathbf{z}_i \sim p(\mathbf{z}), \mathbf{w}_i = f(\mathbf{z}_i)} \left[ d_{\text{geo}}^{k}\left(T_{\mathbf{w}_1}\mathcal{W}_{\mathbf{w}_1}^{k}, T_{\mathbf{w}_2}\mathcal{W}_{\mathbf{w}_2}^{k}\right) \text{ for } k = \min(k_1, k_2) \right], \tag{4}$$

$$I_{local} = \mathbb{E}_{\mathbf{z}_1 \sim p(\mathbf{z}), |\mathbf{z}_2 - \mathbf{z}_1| = \epsilon} \left[ d_{\text{geo}}^{k}\left(T_{\mathbf{w}_1}\mathcal{W}_{\mathbf{w}_1}^{k}, T_{\mathbf{w}_2}\mathcal{W}_{\mathbf{w}_2}^{k}\right) \text{ for } k = \min(k_1, k_2) \right]. \tag{5}$$

where $k_i$ denotes the local dimension estimate at $\mathbf{w}_i = f(\mathbf{z}_i)$. Interestingly, although Distortion metric does not exploit any semantic information, Distortion metric provides a strong correlation between the supervised disentanglement score and the global-basis-compatibility (Choi et al., 2022a).

## 3   FRÉCHET MEAN GLOBAL BASIS

In this section, we propose an unsupervised method for finding global linear perturbation directions that make the same semantic manipulation on the entire latent space, called **Fréchet basis**. If we have such global meaningful perturbations, the vector space representation of latent space along these

semantic basis provides the global semantic representation of a model. The proposed scheme is based on finding the Fréchet mean (Fréchet, 1948; Karcher, 1977) of the local disentangled perturbations. The scheme is in two steps. First, the optimal subspace representing the global semantics is discovered by the Fréchet mean in the Grassmannian manifold (Boothby, 1986) of intrinsic tangent spaces (Choi et al., 2022b) on the target latent space. Second, the optimal basis is obtained as the Fréchet mean in the Special Orthogonal Group (Lang, 2012) of the projected local disentangled perturbations.

## 3.1 METHOD

**Notation**  Throughout this work, we follow the notation presented in Sec 2. Let $\tilde{\mathcal{W}} = \mathbb{R}^{d_{\tilde{\mathcal{W}}}}$ be a ambient target latent space where we want to find global semantic perturbations. We analyze the learned latent manifold $\mathcal{W} = f(\mathcal{Z}) \subset \tilde{\mathcal{W}}$ embedded in this latent space, which is given as an image of the subnetwork from the input noise $\mathcal{Z}$ to $\tilde{\mathcal{W}}$.

**Motivation**  We investigate the problem of discovering global semantic perturbations through the local geometry of learned latent manifold $\mathcal{W}$. Recently, Choi et al. (2022a) discovered that the intrinsic tangent space $T_{\mathbf{w}} \mathcal{W}_{\mathbf{w}}^k$ at each $\mathbf{w} \in \mathcal{W}$ represents the local semantic variation of the generated image from $\mathbf{w}$. Specifically, the intrinsic local dimension at $\mathbf{w}$, denoted as $k$ in $T_{\mathbf{w}} \mathcal{W}_{\mathbf{w}}^k$, corresponds to the number of local semantic perturbations. The top-$k$ components of Local Basis Choi et al. (2022b) are these local semantic perturbations and are the basis vectors of $T_{\mathbf{w}} \mathcal{W}_{\mathbf{w}}^k$ (Eq 3). Hence, the intrinsic tangent space $T_{\mathbf{w}} \mathcal{W}_{\mathbf{w}}^k$ describes the local semantic variation of an image because it is spanned by the local meaningful perturbations.

In this regard, we interpret the global semantic variation as *the mean of these local semantic variations*. One of the most popular methods for defining the mean of subspaces is through *Fréchet mean* (Marrinan et al., 2014). Fréchet mean is a generalization of the mean in vector space to the general metric space. The mean of vectors is the minimizer of the sum of squared distances to each vector. Similarly, Fréchet mean $\mu_{fr}$ in the metric space $X$ with metric $d$ is defined as the minimizer of squared metrics, i.e., for $x_1, x_2, \ldots, x_n \in X$,

$$\mu_{fr} = \arg\min_{\mu \in X} \sum_{1 \le i \le n} d(\mu, x_i)^2. \tag{6}$$

In particular, a Riemannian manifold is an example of metric space where we can introduce the Fréchet mean (Lou et al., 2020). In this work, we utilize the Grassmannian manifold (Boothby, 1986) to find the subspace of latent space for the global semantic representation and the Special Orthogonal Group (Lang, 2012) to choose the optimal basis on it.

### 3.1.1 GLOBAL SEMANTIC SUBSPACE

Our goal is to find a Riemannian manifold where we can embed these intrinsic tangent spaces $\{T_{\mathbf{w}_i} \mathcal{W}_{\mathbf{w}_i}^{k_i}\}_{1 \le i \le n}$ describing local semantic variations at each $\mathbf{w}_i$. The Grassmannian manifold $Gr(k, V)$ denotes the set of $k$-dimensional linear subspaces of vector space $V$ (Boothby, 1986). Hence, *these tangent spaces can be embedded to one Grassmannian manifold $Gr(d_{\mathcal{W}}, \mathbb{R}^{d_{\tilde{w}}})$ by matching their dimensions* to the dimension of learned latent manifold $d_{\mathcal{W}}$. Specifically, we match the dimensions of tangent spaces by refining or extending them to the subspaces spanned by the top-$d_{\mathcal{W}}$ components of Local Basis (Eq 3). This is equivalent to approximating $\mathcal{W}$ with the $d_{\mathcal{W}}$-dimensional local estimate $\mathcal{W}_{\mathbf{w}}^{d_{\mathcal{W}}}$ at all $\mathbf{w} \in \mathcal{W}$ (Eq 2). We estimate the layer-wise dimension $d_{\mathcal{W}}$ of learned latent manifold $\mathcal{W}$ by averaging local dimensions $\{k_i\}_{1 \le i \le n}$ of $n$ i.i.d. samples,

$$T_{\mathbf{w}_i} \mathcal{W}_{\mathbf{w}_i}^{d_{\mathcal{W}}} = \text{span}\{\mathbf{v}_i^{\mathbf{w}_i} : 1 \le i \le d_{\mathcal{W}}\} \in Gr(d_{\mathcal{W}}, \mathbb{R}^{d_{\tilde{w}}}), \tag{7}$$

where $\mathbf{v}_i^{\mathbf{w}_i}$ denotes Local Basis at $\mathbf{w}_i$ (Eq 1). Then, we define the **global semantic subspace** $\mathcal{S}_s$ of $\mathcal{W}$ as the Fréchet mean on $Gr(d_{\mathcal{W}}, \mathbb{R}^{d_{\tilde{w}}})$ with the geodesic metric $d_{geo}$ (Ye & Lim, 2016):

$$\mathcal{S}_s = \arg\min_{\mu \in Gr(d_{\mathcal{W}}, \mathbb{R}^{d_{\tilde{w}}})} \sum_{1 \le i \le n} d_{geo}(\mu, T_{\mathbf{w}_i} \mathcal{W}_{\mathbf{w}_i}^{d_{\mathcal{W}}})^2. \tag{8}$$

Here, the geodesic metric $d_{geo}$ is defined as $d_{\text{geo}}(W, W') = \left(\sum_{i=1}^{k} \theta_i^2\right)^{1/2}$ for $W, W' \in Gr(k, \mathbb{R}^n)$ where $\theta_i$ denotes the $i$-th principal angle between $W$ and $W'$. That is, $\theta_i = \cos^{-1}(\sigma_i(M_W^\top M_{W'}))$

where $M_W \in \mathbb{R}^{n \times k}$ denotes the column-wise concatenation of orthonormal basis for $W$. For optimization, we used the gradient descent algorithm in the Pymanopt (Townsend et al., 2016).

### 3.1.2 Global Semantic Basis

The aim of this work is to find global meaningful perturbations that represent the disentangled semantics. However, the global semantic subspace $\mathcal{S}_s$ is discovered by solving an optimization problem in the Grassmannian manifold, the set of subspaces. We need an additional step to find a specific basis on $\mathcal{S}_s$. In particular, we utilize the Fréchet mean on the Special Orthogonal Group.

**Why Special Orthogonal Group**  Let the columns of $M_{\mathcal{S}}, M'_{\mathcal{S}} \in \mathbb{R}^{d_{\bar{w}} \times d_{\mathcal{W}}}$ be the two distinct orthonormal basis of $\mathcal{S}_s$. Then, there exists an orthogonal matrix $O \in \mathbb{R}^{d_{\mathcal{W}} \times d_{\mathcal{W}}}$, i.e., $O^\top O = OO^\top = I$, which satisfies $M'_{\mathcal{S}} = M_{\mathcal{S}} O$. Therefore, *finding the optimal basis of $\mathcal{S}_s$ is equivalent to finding the orthogonal matrix $O$ given the initial $M_{\mathcal{S}}$*. The Special Orthogonal Group $SO(n)$ consists of $n \times n$ orthogonal matrices with determinant $+1$ (Lang, 2012). Note that the determinant of an arbitrary orthogonal matrix is $+1$ or $-1$. We consider $SO(n)$ instead of the orthogonal matrices for two reasons. First, our task is independent of the flipping (($-1$)-multiplication) of each basis component. The positive perturbation along $v$ is identical to the negative perturbation along $-v$. The flipping of a basis component in $M'_{\mathcal{S}}$ leads to the flipping of the corresponding column in $O$. This results in the ($-1$)-multiplication at the determinant of $O$. Therefore, without loss of generality, we may assume that $O$ is a special orthogonal matrix. Second, the Orthogonal Group is disconnected while the Special Orthogonal Group is connected. Hence, the Orthogonal Group is inadequate for finding the Fréchet mean, which is optimized by the gradient descent algorithm.

**Basis Refinement**  We propose the optimization scheme for finding the global semantic basis from the global semantic subspace $\mathcal{S}_s$. Here, we denote the column-wise concatenation of local semantic basis at each $\mathbf{w}_i$ as $M_{\mathbf{w}_i} \in \mathbb{R}^{d_{\bar{w}} \times d_{\mathcal{W}}}$, i.e., each column is the top-$d_{\mathcal{W}}$ Local Basis at $\mathbf{w} \in \mathcal{W}$. Note that the column space of $M_{\mathbf{w}_i}$ is the local semantic subspace $T_{\mathbf{w}_i} \mathcal{W}^{d_{\mathcal{W}}}_{\mathbf{w}_i}$. Likewise, $M_{\mathcal{S}}$ refers to an initial orthonormal basis of $\mathcal{S}_s$. As an overview, the proposed scheme is as follows:

  (i) Project each local semantic basis to the $d_{\mathcal{W}}$-dimensional global semantic subspace $\mathcal{S}_s$.

 (ii) Project these projected local semantic basis to $SO(d_{\mathcal{W}})$.

(iii) Find the Fréchet mean $O$ in $SO(d_{\mathcal{W}})$ and embed $O$ back to the ambient space $\mathbb{R}^{d_{\bar{w}}}$.

Before the above optimization, we preprocess each local semantic basis at $\mathbf{w}_i$, i.e., the columns of $M_{\mathbf{w}_i}$, to be positively aligned to each column of $M_{\mathcal{S}}$, i.e., $\langle M_{\mathbf{w}_i}[:, i], M_{\mathcal{S}}[:, i] \rangle > 0$ for all $i$. (i) As a first step, we project each local semantic basis at $\mathbf{w}_i$ onto the global semantic subspace $\mathcal{S}_s$, i.e., $M_{\mathcal{S}}^\top M_{\mathbf{w}_i}$. (ii) Then, the matrix of projected local semantic basis $M_{\mathcal{S}}^\top M_{\mathbf{w}_i} \in \mathbb{R}^{d_{\mathcal{W}} \times d_{\mathcal{W}}}$ is projected to $SO(d_{\mathcal{W}})$[1]. The projection on the orthogonal group $P_o$ and the proposed projection on the Special Orthogonal Group $P_{so}$ can be obtained via SVD. (See the appendix for proof.): Let $X = U\Sigma V^\top$ be a SVD of $X \in \mathbb{R}^{d_{\mathcal{W}} \times d_{\mathcal{W}}}$,

$$P_{so}(X) = U \operatorname{diag}\left(1, 1, \ldots, 1, \det\left(P_o(X)\right)\right) V^\top \quad \text{where} \quad P_o(X) = UV^\top. \tag{9}$$

(iii) Finally, we find the optimal orthogonal matrix $O$, that transforms the initial basis $M_{\mathcal{S}}$ to the **global semantic basis** $\mathcal{B}_s$, via Fréchet mean of projected local semantic basis $\{P_{so}\left(M_{\mathcal{S}}^\top M_{\mathbf{w}_i}\right)\}_i \subset SO(d_{\mathcal{W}})$.

$$\mathcal{B}_s = M_{\mathcal{S}} O \quad \text{where} \quad O = \operatorname*{arg\,min}_{\mu \in SO(d_{\mathcal{W}})} \sum_{1 \leq i \leq n} d\left(\mu, P_{so}\left(M_{\mathcal{S}}^\top M_{\mathbf{w}_i}\right)\right)^2. \tag{10}$$

Here, the Riemannian metric on $SO(d_{\mathcal{W}})$ is defined as $d(X_1, X_2) = \| \operatorname{Skew}\left(\log(X_1^\top X_2)\right) \|_F$ for $X_1, X_2 \in SO(d_{\mathcal{W}})$ where $\log$ denotes a matrix logarithm and Skew refers to the skew-symmetric component of a matrix, i.e., $\operatorname{Skew}(X_1) = \frac{1}{2}\left(X_1 - X_1^\top\right)$. As in the Grassmannian manifold, we utilized the Pymanopt (Townsend et al., 2016) implementation for finding the Fréchet mean on $SO(d_{\mathcal{W}})$. Then, the global semantic basis $\mathcal{B}_s$ is obtained as the embedding of Fréchet mean $O$ to $\mathbb{R}^{d_{\bar{w}}}$, i.e., $\mathcal{B}_s = M_{\mathcal{S}} O$. We call this global semantic basis $\mathcal{B}_s$ **Fréchet Basis**.

---

[1]This projection is underdetermined for the matrix with determinant 0 because of the subspace generated by singular vectors with $\sigma = 0$. Because these matrixes are measure-zero set, this did not happen in practice.

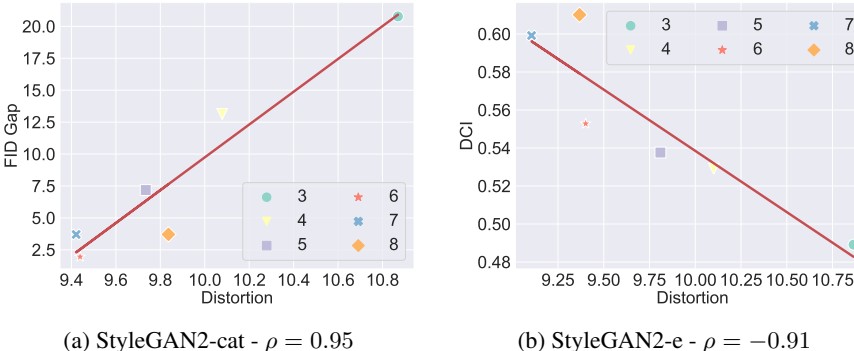

(a) StyleGAN2-cat - $\rho = 0.95$          (b) StyleGAN2-e - $\rho = -0.91$

Figure 2: **Correlations of $L^2$-Distortion ($\downarrow$) to FID gap ($\downarrow$) and DCI ($\uparrow$)** when $\theta_{pre} = 0.005$. $L^1$-Distortion shows the correlations of 0.98 to FID-Gap in StyleGAN2-cat and of -0.91 to DCI in StyleGAN2-e. (See the appendix for full correlation results in six models.)

## 3.2   FRÉCHET BASIS AS $L^2$-DISTORTION MINIMIZER

The Fréchet basis can be interpreted as the minimizer of the unsupervised disentanglement metric, Distortion (Choi et al., 2022a). Distortion metric is based on the inconsistency of intrinsic tangent spaces. Specifically, Distortion metric $\mathcal{D}_{\mathcal{W}}$ is the ratio between the inconsistency at two random $\mathbf{w} \in \mathcal{W}$ and at two close $\mathbf{w} \in \mathcal{W}$, i.e., $\mathcal{D}_{\mathcal{W}} = I_{rand}/I_{local}$ (Eq 4 and 5). Here, the intrinsic tangent space represents the local semantic variations. From this point of view, the Distortion-based global semantic subspace $\mathcal{S}_{\mathcal{D}}$ would be a representative of these tangent spaces that minimize the inconsistency to each tangent space.

$$\mathcal{S}_{\mathcal{D}} = \arg\min_{\mu \leq \mathbb{R}^{d_{\bar{\mathcal{W}}}}} I_{global}(\mu) \quad \text{with} \quad I_{global}(\mu) = \mathbb{E}_{\mathbf{z} \sim p(\mathbf{z}), \, \mathbf{w} = f(\mathbf{z})} \left[ d_{\text{geo}}^k \left( \mu^k, T_{\mathbf{w}} \mathcal{W}_{\mathbf{w}}^k \right) \right], \quad (11)$$

where $\mu$ is a subspace of $\mathbb{R}^{d_{\bar{w}}}$, $k$ refers to the local dimension at $\mathbf{w}$ and $\mu^k$ denotes the $k$-dimensional refinement of $\mu$. The Fréchet basis assumes that the entire latent manifold $\mathcal{W}$ is approximated with $d_{\mathcal{W}}$-dimensional local estimate at all $\mathbf{w} \in \mathcal{W}$. Under this assumption, $I_{global}(\mu)$ becomes

$$I_{global}(\mu) = \left( 1/\sqrt{d_{\mathcal{W}}} \right) \cdot \mathbb{E}_{\mathbf{z} \sim p(\mathbf{z}), \, \mathbf{w} = f(\mathbf{z})} \left[ d_{\text{geo}} \left( \mu, T_{\mathbf{w}} \mathcal{W}_{\mathbf{w}}^{d_{\mathcal{W}}} \right) \right] \quad \text{for } \mu \in Gr(d_{\mathcal{W}}, \mathbb{R}^{d_{\bar{w}}}). \quad (12)$$

The comparison with Eq 8 shows that the global semantic subspace by Fréchet mean $\mathcal{S}_s$ can be interpreted as $L^2$-Distortion minimizer, i.e., $d_{geo}^2$ instead of $d_{geo}$. Although the original $L^1$-Distortion was proven to provide high correlations with the global-basis-compatibility and the supervised disentanglement score, $L^2$-Distortion was not tested (Choi et al., 2022a). Therefore, we evaluated whether $L^2$-Distortion is also a meaningful metric to verify the validity of minimizing it. Following the experiments in Choi et al. (2022a), we assessed the global-basis-compatibility by the FID (Heusel et al., 2017) Gap between Local Basis and GANSpace (Härkönen et al., 2020) under the same perturbation intensity. Also, DCI score (Eastwood & Williams, 2018) is adopted as the supervised disentanglement score. We utilized 40 binary attribute classifiers pre-trained on CelebA (Liu et al., 2015) to annotate the 10k generated images. Figure 2 demonstrates that $L^2$-Distortion achieves high correlations comparable to the original Distortion score in the global-based compatibility and DCI. These results prove that our framework of minimizing $L^2$-Distortion by Fréchet mean is also valid.

## 4   EXPERIMENTS

### 4.1   FRÉCHET BASIS AS GLOBAL SEMANTIC PERTURBATIONS

We evaluate the Fréchet basis as the global semantic perturbations on the intermediate layers of the mapping network in various StyleGAN models (Karras et al., 2019; 2020b). For each StyleGAN model, we used the layers from 3rd to 8th because the local dimension estimate is rather unstable for the 1st and 2nd layers depending on the preprocessing hyperparameter $\theta_{pre}$ (Choi et al., 2022a). We chose these intermediate layers for evaluation because they are diverse and properly disentangled latent spaces. In this manner, the Fréchet basis can be tested on six latent spaces for each pre-trained

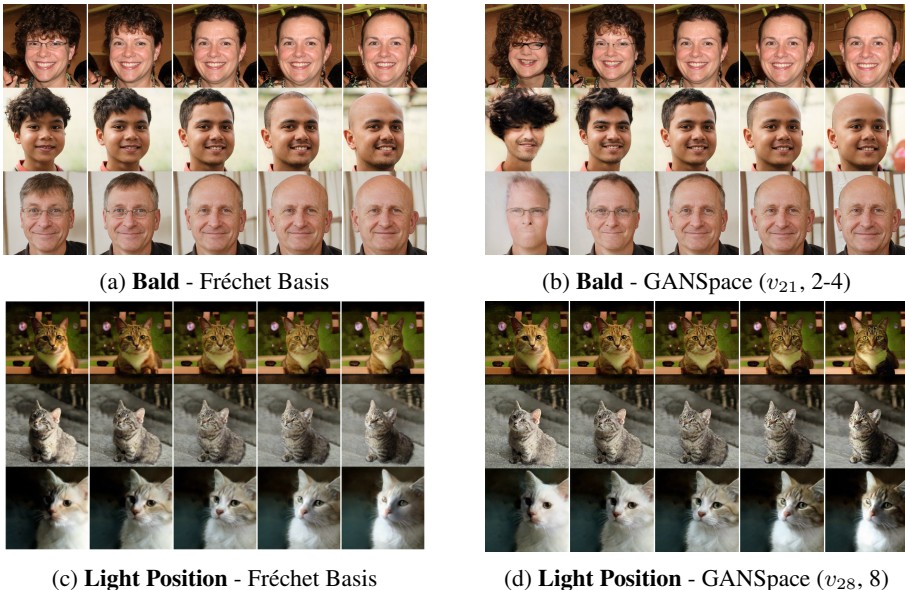

(a) **Bald** - Fréchet Basis

(b) **Bald** - GANSpace ($v_{21}$, 2-4)

(c) **Light Position** - Fréchet Basis

(d) **Light Position** - GANSpace ($v_{28}$, 8)

Figure 3: **Comparison of Semantic Factorization** between Fréchet basis and GANSpace. ($v_i$, $l_1$-$l_2$) denotes the layer-wise edit along the $i$-th GANSpace component at the $l1$-$l2$ layers. The image traversals are performed on StyleGAN2-FFHQ (Fig 3a, 3b) and StyleGAN2-LSUN cat (Fig 3c, 3d).

**StyleGAN model.** In this section, all Fréchet basis are discovered using 1,000 i.i.d. samples of Local Basis with $\theta_{pre} = 0.01$. The max iteration is set to 200 when optimizing Fréchet mean with Pymanopt. The evaluation is performed on two properties: Semantic Factorization and Robustness. Fréchet basis is compared with GANSpace (Härkönen et al., 2020) and SeFa (Shen & Zhou, 2021) because these two methods are also unsupervised global basis (Sec 2). Note that GANSpace and Fréchet basis can be applied to arbitrary latent space, but SeFa is only applicable to $\mathcal{W}$-space (Karras et al., 2019), i.e., the last layer of the mapping network in StyleGANs.

**Semantic Factorization** In Fig 3 and 4, we evaluated the semantic factorization of Fréchet basis. Figure 3 shows how the image changes as we perturb the latent variable along each global basis. For a fair comparison, we took the annotated basis in GANSpace (Härkönen et al., 2020) and compared those with Fréchet basis on $\mathcal{W}$-space of three StyleGAN models. Because GANSpace performs layer-wise edits, we matched the set of layers, where the perturbed latent variable is fed, in the synthesis network as annotated. The corresponding Fréchet basis component is selected by the cosine-similarity. Each subfigure shows the three images traversed with the same global basis, perturbation intensity, and the set of perturbed layers. The original image is placed at the center. Hence, these subfigures also show the semantic consistency of the global basis. In StyleGAN2 trained on FFHQ, GANSpace shows image failure on the left side and semantic inconsistency on the third row (not representing hairy on the left) (Fig 3b). In StyleGAN2 trained on LSUN-cat (Yu et al., 2015), GANSpace presents entangled semantic manipulation (Fig 3d). The latent traversal along GANSpace changes the light position as annotated, but also darkens the striped pattern of cats. On the other hand, Fréchet basis achieves better semantic factorization without showing those problems (Fig 3a and 3c). (See the appendix G for additional examples of other attributes and datasets. Also, since Fréchet basis is an average of Local Basis, we compared these two methods and GANSpace in the appendix F.)

For the quantitative comparison of semantic factorization, we compared DCI as in Sec 3.2. DCI is a supervised disentanglement metric that assesses the *axis-wise alignment* of semantics. Hence, we measured DCIs of the latent space representations with two global basis, GANSpace and Fréchet basis. Specifically, we converted a latent variable $\mathbf{w} \in \mathbb{R}^{d_{\bar{w}}}$ into the $d_{\mathcal{W}}$-dimensional representation with each global basis: For a $d_{\mathcal{W}}$-dimensional global basis $\{\mathbf{v}_i^{global}\}_{1 \leq i \leq d_{\mathcal{W}}}$,

$$\text{Representation of } \mathbf{w} \text{ with } \{\mathbf{v}_i^{global}\}_{1 \leq i \leq d_{\mathcal{W}}} = \left( \mathbf{w} \cdot \mathbf{v}_1^{global}, \mathbf{w} \cdot \mathbf{v}_2^{global}, \ldots, \mathbf{w} \cdot \mathbf{v}_{d_{\mathcal{W}}}^{global} \right). \quad (13)$$

Figure 4 presents the DCI results. StyleGAN2-e denotes the config-e model of StyleGAN2 (Karras et al., 2020b), and all three models are trained on FFHQ (Karras et al., 2019). The intrinsic dimension

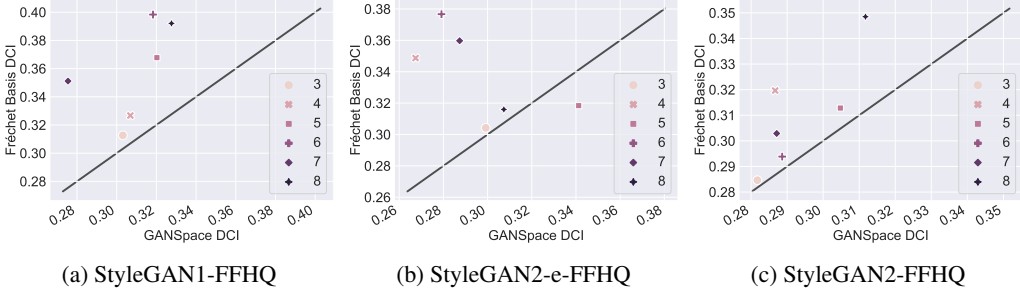

(a) StyleGAN1-FFHQ  (b) StyleGAN2-e-FFHQ  (c) StyleGAN2-FFHQ

Figure 4: **Quantitative Comparison of Semantic Factorization.** DCI score ($\uparrow$) is a supervised disentanglement metric that evaluates the axis-wise alignment of semantics. Fréchet basis outperforms GANSpace when a point is located *above* the black line. (The black line illustrates $y = x$.)

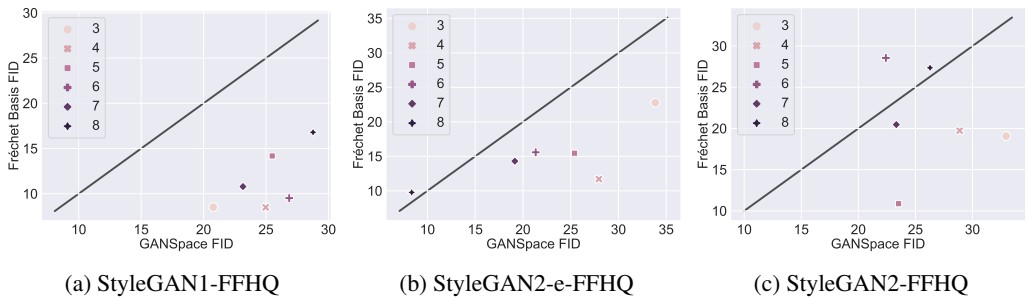

(a) StyleGAN1-FFHQ  (b) StyleGAN2-e-FFHQ  (c) StyleGAN2-FFHQ

Figure 5: **Quantitative Comparison of Robustness.** The image fidelity under the latent traversal is evaluated by FID ($\downarrow$). Fréchet basis performs better when a point is placed *below* the black line.

of each latent space is estimated with $\theta_{pre} = 0.01$ (Choi et al., 2022a). In all latent spaces except for the 5-th layer in StyleGAN2-e, the latent space achieves a higher DCI score when represented with Fréchet basis. This quantitative result shows that Fréchet basis provides a better semantic factorization along each basis component at the *same latent space*.

**Robustness**    We tested the robustness of Fréchet basis by comparing the image fidelity under the latent perturbation. For each global basis, we evaluated FID (Heusel et al., 2017) of 50k i.i.d. latent perturbed images. The perturbation direction is selected to be the 1st component for the GANSpace. In Fréchet basis, we chose the component with the highest cosine-similarity to the 1st component of GANSpace. The perturbation intensity is 2 in StyleGAN1, and 5 in StyleGAN2-e and StyleGAN2. The FID scores of the latent spaces in three models are provided in Fig 5. (See the appendix for FID scores under various perturbations intensity.) We think this higher robustness is because the global semantic subspace $\mathcal{S}_s$ is the Fréchet mean of the intrinsic tangent spaces of learned latent manifold. The strong robustness of traversing along the tangent space at each latent variable was observed in Choi et al. (2022b). Therefore, traversing along Fréchet basis can be interpreted as traversing along the mean of these locally robust perturbation directions, because Fréchet basis is a basis of $\mathcal{S}_s$.

## 4.2 BASIS REFINEMENT BY FRÉCHET MEAN

Fréchet basis consists of two parts: finding the *global semantic subspace* $\mathcal{S}_s$ and selecting the *global semantic basis* $\mathcal{B}_s$ from the discovered semantic subspace. In particular, the second step can be utilized to refine a given global basis from previous methods. We ran this basis refinement on the subspace generated by the existing global methods, GANSpace and SeFa. This experiment reveals the contribution of the second step.

Table 1: **Basis Refinement by Fréchet mean**.

| Global Basis | FID ($\downarrow$) | DCI ($\uparrow$) |
|---|---|---|
| Fréchet basis | **7.49** | **0.350** |
| GANSpace | 8.90 | 0.312 |
| GANSpace-refine | 8.63 | 0.336 |
| SeFa | 11.79 | 0.307 |
| SeFa-refine | 7.89 | 0.337 |

Let $\mathcal{S}_{global}$ be the $d_{\mathcal{W}}$-dimensional semantic subspace generated by the global basis $\{\mathbf{v}_i^{global}\}_{1 \le i \le d_{\mathcal{W}}}$. Then, we optimized the constrained-optimal global basis

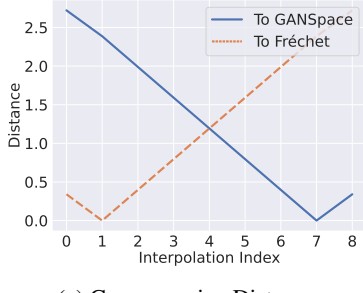
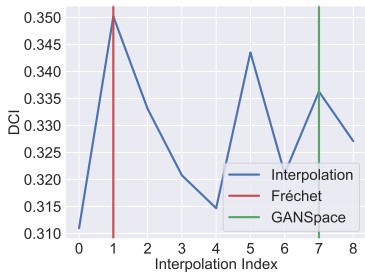

(a) Grassmannian Distance

(b) DCI score of Interpolation basis $\mathcal{B}_i$

Figure 6: **Geodesic Interpolation** from Fréchet basis ($i = 1$) to GANSpace ($i = 7$).

$\mathcal{B}_{global}$ in $\mathcal{S}_{global}$ via Sec 3.1.2. Table 1 shows FID and DCI scores of Fréchet basis, GANSpace, GANSpace refinement, SeFA, and SeFa refinement in $\mathcal{W}$-space of StyleGAN2 trained on FFHQ. These two scores are evaluated in the same manner as in Sec 4.1 with the perturbation intensity 2 for FID. Most importantly, Fréchet basis achieves the best FID and DCI. Also, the basis refinement monotonically improves the two scores of the two previous methods, GANSpace and SeFa. The comparison with the previous global method and its refinement proves the contribution of basis optimization. Also, the superior performance of Fréchet basis over the two refinements shows the contribution of subspace optimization.

## 4.3 GEODESIC INTERPOLATION ON GRASSMANNIAN MANIFOLD

We further investigate the optimality of global semantic subspace $\mathcal{S}_s$ by analyzing the interpolation from $\mathcal{S}_s$ to GANSpace subspace $\mathcal{S}_{GS}$. Similar to Sec 4.2, this experiment examines the contribution of the first step in Fréchet basis. In the Grassmannian manifold $Gr(k, \mathbb{R}^n)$, there exists at least one length-minimizing geodesic between any two subspaces in $Gr(k, \mathbb{R}^n)$. Moreover, there is an explicit parametrization of this length-minimizing geodesic (Eq 14) (Chakraborty & Vemuri, 2015). For $\mathcal{X}, \mathcal{Y} \in Gr(k, \mathbb{R}^n)$, let $X, Y \in \mathbb{R}^{n \times k}$ be the column-wise concatenation of their orthonormal basis. Then, the length-minimizing geodesic $\Gamma(\mathcal{X}, \mathcal{Y}, t)$ from $\mathcal{X}$ to $\mathcal{Y}$ is defined as:

$$\Gamma(\mathcal{X}, \mathcal{Y}, t) = \text{span}\{ (XV \cos(\Theta t) + U \sin(\Theta t)) V^\top \} \quad \text{for } t \in [0, 1], \tag{14}$$

where $X^\top Y$ is non-singular, $(Y - XX^\top Y)(X^\top Y)^{-1} = U\Sigma V^\top$ is the thin SVD, and $\Theta = \arctan \Sigma$. Note that the last $V^\top$ multiplication is unnecessary as the subspace. However, we added it to match the basis to $X$ at $t = 0$. We used this geodesic to perform the interpolation and extrapolation from $\mathcal{S}_s$ to $\mathcal{S}_{GS}$:

$$\mathcal{S}_i = \Gamma(\mathcal{X}, \mathcal{Y}, (i - 1)/n) \quad \text{for } i = 0, 1, \ldots, n + 2. \tag{15}$$

with $n = 6$. Note that $i = 0$, $n + 2$ represent the extrapolations of $t = (-1/n)$, $1 + (1/n)$. Because the above interpolation is performed on a subspace-scale, we conducted the basis refinement (Sec 4.2) for each interpolation subspace $\mathcal{S}_i$ to find the interpolation basis $\mathcal{B}_i$. Figure 6 presents the geodesic metric $d_{geo}$ in the Grassmannian manifold for each interpolation subspace to GANSpace and Fréchet basis (Fig 6a), and the DCI score evaluated at each interpolation basis $\mathcal{B}_i$ (Fig 6b) on $\mathcal{W}$-space of StyleGAN2-FFHQ. First, Fig 6a shows that $\mathcal{S}_i$ performs interpolation from Fréchet basis at $i = 0$ to GANSpace at $i = 7$. This interpolation is linear in the Grassmannian metric $d_{geo}$. Second, the DCI score at each interpolation basis $\mathcal{B}_i$ are presented in Fig 6b. Fréchet basis at $i = 0$ achieves the best DCI score among the interpolation basis. Note that the DCI score of original GANSpace without refinement is 0.312 (Tab 1), which is lower than the score after refinement at $i = 7$ as in Sec 4.2.

## 5 CONCLUSION

In this paper, we proposed the unsupervised global semantic basis on the intermediate latent space in a GAN, called Fréchet basis. Fréchet basis is discovered by utilizing the Fréchet mean on the Grassmannian manifold and the Special Orthogonal Group. Our experiments demonstrate that Fréchet basis achieves better semantic factorization and robustness than the previous unsupervised global methods. In addition, we suggest the basis refinement scheme using the Fréchet mean. Given the same semantic subspace generated by the previous global methods, the refined basis attains better semantic factorization and robustness.

## ACKNOWLEDGMENTS

This work was supported by a KIAS Individual Grant [AP087501] via the Center for AI and Natural Sciences at Korea Institute for Advanced Study, the NRF grant [2012R1A2C3010887], and the MSIT/IITP ([1711117093], [2021-0-00077], [No. 2021-0-01343, Artificial Intelligence Graduate School Program(SNU)]).

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

## A    PROJECTION ONTO SPECIAL ORTHOGONAL GROUP

In this section, we provide proof for the projection of an invertible matrix $A \in GL(n)$ onto the special orthogonal group $SO(n)$. Formally, the set of invertible matrices $GL(n)$, orthogonal group $O(n)$, and special orthogonal group $SO(n)$ are defined as follows:

$$GL(n) := \{A \in \mathbb{R}^{n \times n} : \ \det(A) \neq 0\}, \tag{16}$$

$$O(n) := \{A \in \mathbb{R}^{n \times n} : \ A^\top A = AA^\top = I\}. \tag{17}$$

$$SO(n) := \{A \in \mathbb{R}^{n \times n} : \ A^\top A = AA^\top = I, \det(A) = 1\}. \tag{18}$$

For a non-invertible matrix, the projection onto $SO(n)$ is not uniquely defined because of the subspace generated by singular vectors with $\sigma = 0$. However, the set of non-invertible matrices has measure-zero in the set of $n \times n$ matrices $\mathbb{R}^{n \times n}$. Thus, this did not happen in practice during the Fréchet basis optimization.

**Theorem 1.** *The following optimization problem, i.e., the projection of $A \in GL(n)$ onto the Special Orthogonal Group $SO(n)$,*

$$\underset{X \in SO(n)}{\arg \min} \ \|X - A\|_F \quad where \ A \in GL(n), \tag{19}$$

*has a solution*

$$P_{so}(A) = X^* = U \ \mathrm{diag}\left(1, 1, \ldots, 1, \det\left(UV^\top\right)\right) \ V^\top \tag{20}$$

*where $A = U\Sigma V^\top$ is the Singular Value Decomposition (SVD) of $A$ with the projection onto orthogonal group $P_o(A) = UV^\top$. (The projection is unique if we assume $\sigma_1 > \sigma_2 > \ldots > \sigma_n > 0$ where $\{\sigma_i\}_{1 \leq i \leq n}$ denote the singular values of $A$.)*

*Proof.* 1. If $P_o(A) = UV^\top \in SO(n)$, done. ($\because SO(n) \subset O(n)$ and $\det\left(P_o(A)\right)) = 1$).

2. If $P_o(A) = UV^\top \notin SO(n)$, i.e., $\det(A) < 0$,

$$\|X - A\|_F^2 = \|I - X^\top A\|_F^2 = n - 2\operatorname{tr}(X^\top A) + \operatorname{tr}(A^\top A). \tag{21}$$

For any skew-symmetric matrix $K$, define $f(t)$ as

$$f(t) = -2\operatorname{tr}(A^\top X e^{tK}), \quad \text{for } t \in \mathbb{R}. \tag{22}$$

Note that $Xe^{tK} \in SO(n)$ and $\operatorname{tr}(A^\top A)$ is given. Therefore, if $X$ is the minimizer of $\|X - A\|_F$, we have $f'(0) = -2\operatorname{tr}(A^\top X K) = 0$ for all skew-symmetric $K$. Thus, $A^\top X$ is symmetric.

Without loss of generality, we may assume that $A = U_0 \Sigma_0 V_0^\top, X = U_0 X' V_0^\top$ for some $U_0, V_0 \in SO(n)$ where $\Sigma_0$ is the diagonal matrix as $\Sigma$ in SVD but $\Sigma_0$ might have the negative elements. This decomposition can be obtained by flipping the singular vectors in $U, V$ of SVD to make $U_0, V_0 \in SO(n)$ and letting $X' = U_0^\top X V_0$. Explicitly, from $\det(A) < 0$,

$$U_0 = U \ \mathrm{diag}\left(1, \ldots, 1, \det\left(U\right)\right), \qquad V_0 = V \ \mathrm{diag}\left(1, \ldots, 1, \det\left(V\right)\right),$$
$$\Sigma_0 = \mathrm{diag}\left(\sigma_1, \ldots, \sigma_{n-1}, -\sigma_n\right). \tag{23}$$

Then, since $X^\top A = \left(A^\top X\right)^\top$,

$$X^\top A = V_0 \left(X'\right)^\top \Sigma_0 V_0^\top \text{ is symmetric and has negative determinant.} \tag{24}$$

Therefore, $\left(X'\right)^\top \Sigma_0$ is also symmetric and thus diagonalizable.

$$\|X - A\|_F^2 = \|I - X^\top A\|_F^2 = \|I - V_0 \left(X'\right)^\top \Sigma_0 V_0^\top\|_F^2$$
$$= \|I - \left(X'\right)^\top \Sigma_0\|_F^2 = \sum_i \left(1 - \lambda_i\right)^2, \tag{25}$$

where $\lambda_i$ denotes the $i$-th eigenvalue of $\left(X'\right)^\top \Sigma_0$ with $|\lambda_1| \geq |\lambda_2| \geq \ldots \geq |\lambda_n|$. Note that since $X' = U_0^\top X V_0 \in SO(n)$, the $i$-th singular value $\sigma_i$ of $A$ satisfies $\sigma_i = |\lambda_i|$. Moreover, an odd number of signed singular values in $\Sigma_0$ is negative because $\det(A) < 0$. Also, $\det(\left(X'\right)^\top \Sigma_0) = \det(X^\top A) < 0$ implies that an odd number of eigenvalues is negative. Hence, $\|X - A\|_F$ is minimized when $\lambda_i = \sigma_i$ for $1 \leq i \leq n - 1$ and $\lambda_n = -\sigma_n$. If $X'$ is the diagonal matrix satisfying this condition, $X = U \ \mathrm{diag}\left(1, 1, \ldots, 1, -1\right) V^\top$. Lemma 1 proves that when $\sigma_1 > \sigma_2 > \ldots > \sigma_n$ is satisfied, the uniqueness of $X'$ is guaranteed. $\qquad \square$

**Lemma 1.** *Let $X' \in SO(n)$ and $\Sigma_0 = \mathrm{diag}\left(\sigma_1, \ldots, \sigma_{n-1}, -\sigma_n\right)$ with $\sigma_1 > \ldots > \sigma_{n-1} > \sigma_n > 0$.*

$$\text{If } Y = (X')^\top \Sigma_0 \text{ is symmetric, } X' \text{ is a diagonal matrix.} \tag{26}$$

*Proof.* Since $Y$ is symmetric, it is diagonalizable with the orthogonal matrix $P \in O(n)$.

$$Y = (X')^\top \Sigma_0 = PDP^t. \tag{27}$$

We interpret these two decompositions $Y = (X')^\top \Sigma_0 I = PDP^t$ as two SVD-like representations of $Y$ because $(X')^\top \in SO(n)$ and $I^\top = I$. The ordered singular vectors in the domain of $Y$ are uniquely determined as the basis for each eigenspace of $Y^\top Y$. Note that if the dimension of eigenspace is bigger than 1, there is a freedom of choosing a basis in it.

From $\Sigma_0$, the possible eigenvalues of $Y$ are $\{\pm\sigma_i\}_{1 \leq i \leq n}$ and the eigenvalues of $Y^\top Y$ are $\{\sigma_i^2\}_{1 \leq i \leq n}$. Therefore, since the standard basis $\{e_i\}_{1 \leq i \leq n}$ of $\mathbb{R}^n$ are the domain singular vectors of $Y$ from $Y = (X')^\top \Sigma_0 I$ and $Y$ is diagonalizable,

$$Y(e_i) = \begin{cases} \sigma_i e_i & \text{for } 1 \leq i \leq n-1 & (\because \sigma_1 > \ldots > \sigma_{n-1} > 0) \\ -\sigma_n e_n & \text{for } i = n & (\because \sigma_{n-1} > \sigma_n > 0). \end{cases} \tag{28}$$

Hence, the standard basis are also the codomain singular vectors of $Y$, which implies that $X'$ is diagonal. $\qquad\square$

## B  IMPLEMENTATION DETAIL

In this section, we summarize the hyperparameters for Frechet basis presented in the experimental results in Section 4.

- Preprocossing hyperparameter $\theta_{pre}$ for local dimension estimation Choi et al. (2022a): $\theta_{pre} = 0.01$.
- Global Semantic Subspace Optimization
  - Number of samples $n = 1000$.
  - Max iteration in Frechet mean Optimization using Pymanopt Townsend et al. (2016) $= 1,000$.
  - Max time in Frechet mean Optimization using Pymanopt $= 2,000$.
- Global Semantic Basis Optimization
  - Number of samples $n = 1000$.
  - Max iteration in Frechet mean Optimization using Pymanopt $= 200$.
  - Max time in Frechet mean Optimization using Pymanopt $= 10,000$.

## C    FULL CORRELATION RESULTS OF $L^2$-DISTORTION

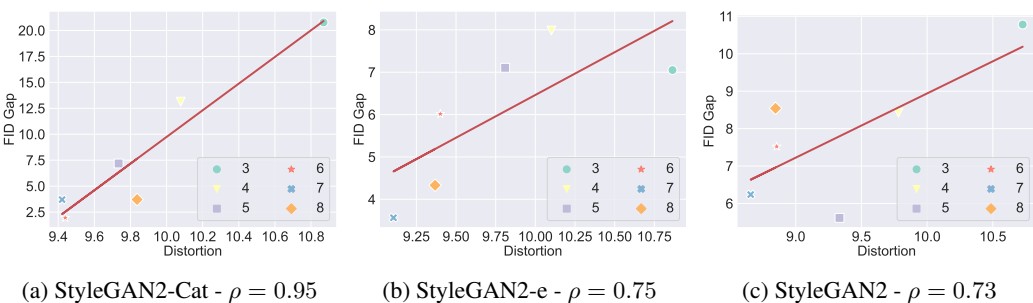

(a) StyleGAN2-Cat - $\rho = 0.95$      (b) StyleGAN2-e - $\rho = 0.75$      (c) StyleGAN2 - $\rho = 0.73$

Figure 7: **Correlation between L2-Distortion metric ($\downarrow$) and FID gap ($\downarrow$)** when $\theta_{pre} = 0.005$. The correlations $\rho$ are 0.98 for StyleGAN2-cat, 0,73 for StyleGAN2-e, and 0.70 for StyleGAN2 in $L^1$-Distortion.

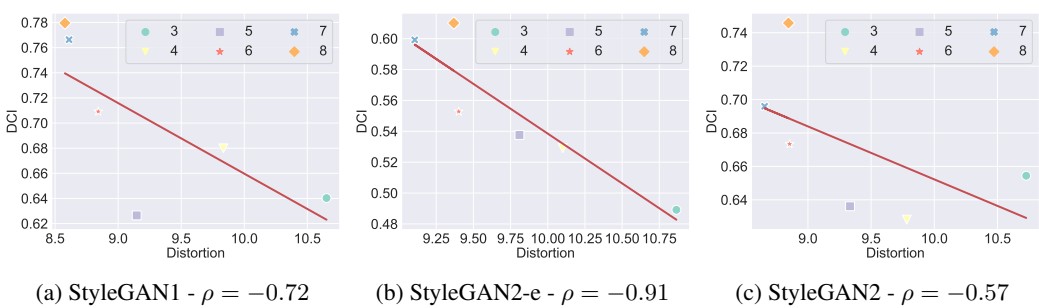

(a) StyleGAN1 - $\rho = -0.72$      (b) StyleGAN2-e - $\rho = -0.91$      (c) StyleGAN2 - $\rho = -0.57$

Figure 8: **Correlation between L2-Distortion metric ($\downarrow$) and DCI ($\uparrow$)** when $\theta_{pre} = 0.005$. The correlations $\rho$ are 0.98, 0,73, and 0.70 for StyleGAN2-cat, StyleGAN2-e, and StyleGAN2 for L1-Distortion.

# D  QUANTITATIVE COMPARISON OF ROBUSTNESS (FID) FOR VARIOUS PERTURBATION INTENSITY

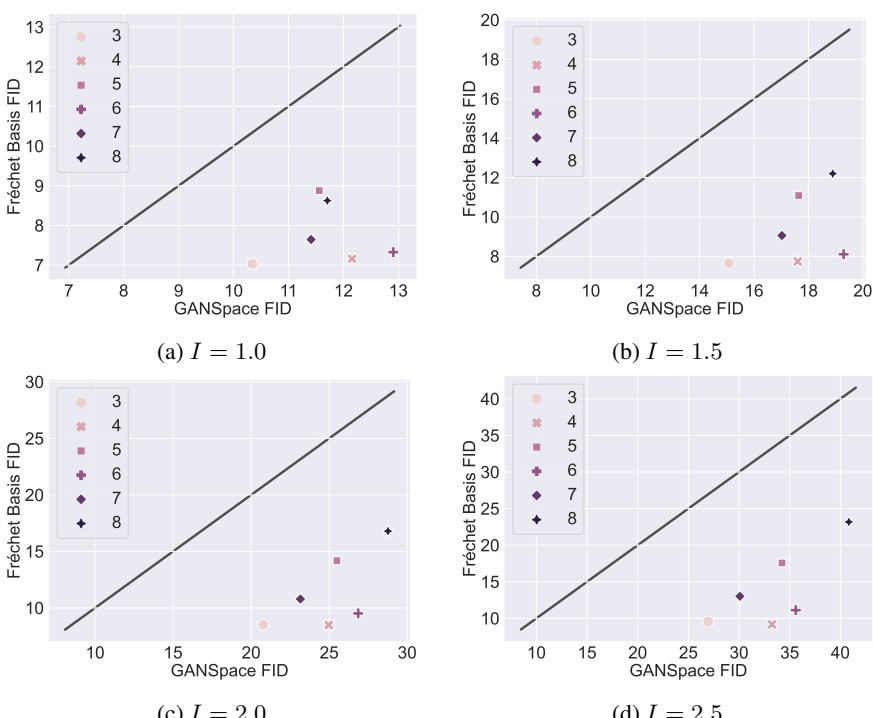

Figure 9: **Quantitative Comparison of Robustness** between GANSpace and Fréchet basis on StylGAN1-FFHQ (Karras et al., 2019) for various perturbation intensity $I$. The perturbation intensity $I$ is measured by the $L^2$-norm on each latent space. The image fidelity under the latent traversal is evaluated by FID (↓). The black line indicates where two FIDs are equal.

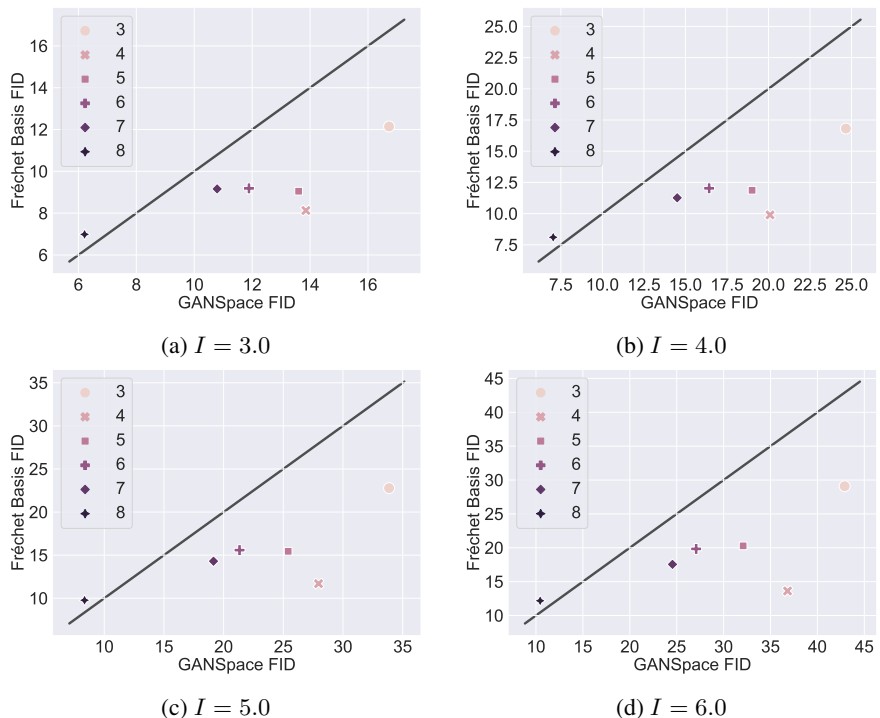

Figure 10: **Quantitative Comparison of Robustness** between GANSpace and Fréchet basis on StylGAN2-e-FFHQ (Karras et al., 2020b) for various perturbation intensity $I$. The image fidelity under the latent traversal is evaluated by FID ($\downarrow$). The black line indicates where two FIDs are equal.

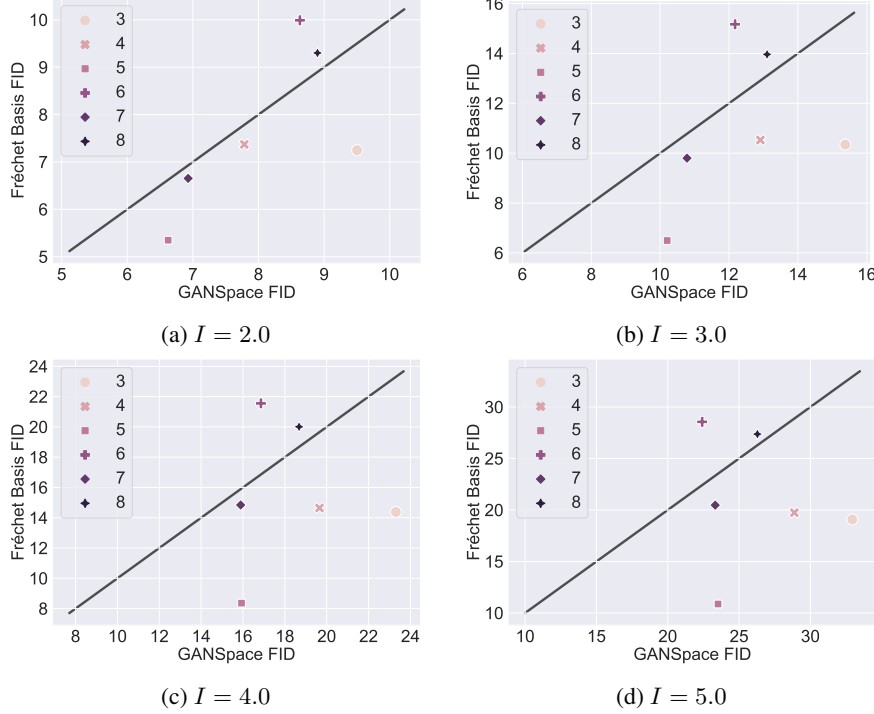

Figure 11: **Quantitative Comparison of Robustness** between GANSpace and Fréchet basis on StylGAN2-FFHQ (Karras et al., 2020b) for various perturbation intensity $I$. The image fidelity under the latent traversal is evaluated by FID ($\downarrow$). The black line indicates where two FIDs are equal.

# E    ABLATION STUDY ON THE OTHER MEANS OF GRASSMANNIAN MANIFOLD

In this section, we conducted an ablation study on defining the global semantic subspace in the Grassmanifold. In particular, we compared the Fréchet mean Fréchet (1948); Marrinan et al. (2014) and extrinsic mean Srivastava & Klassen (2002) of the Grassmannian manifold. The extrinsic mean $\mu_E$ is defined as the minimizer of squared extrinsic metrics $d_E$, i.e., for $x_1, \ldots, x_n \in Gr(k, \mathbb{R}^n)$,

$$\mu_E = \arg\min_{\mu \in X} \sum_{1 \leq i \leq n} d_E\left(\mu, x_i\right)^2, \quad \text{where } d_E\left(\mu, x_i\right) = d_\Phi\left(\Phi(\mu), \Phi(x_i)\right), \quad (29)$$

where $\Phi$ denotes an appropriate embedding of $Gr(k, \mathbb{R}^n)$. Following Marrinan et al. (2014), we set the embedding $\Phi$ to be the corresponding projection $\mathbb{P}_{x_i}$, i.e., $\Phi(x_i) = \mathbb{P}_{x_i} = M_{x_i}^\top M_{x_i}$ where $M_{x_i} \in \mathbb{R}^{n \times k}$ indicates the column-wise concatenation of an orthonormal basis of $x_i$. Also, the Frobenius norm is adopted for $d_\Phi$.

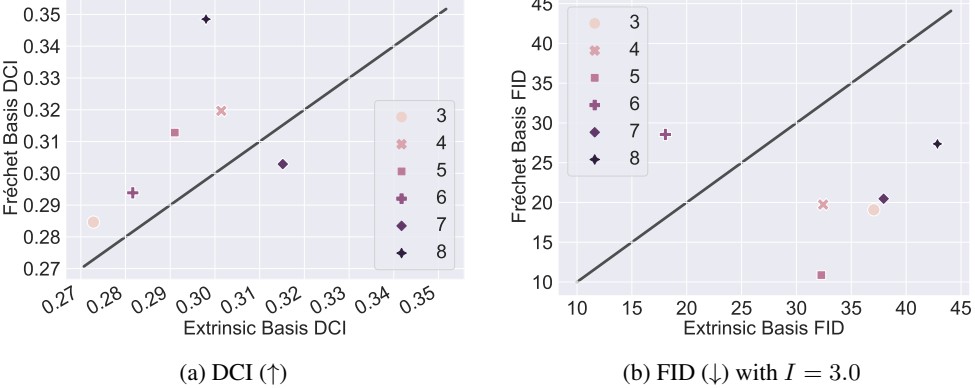

(a) DCI ($\uparrow$)                (b) FID ($\downarrow$) with $I = 3.0$

Figure 12: **Ablation study on the means of Grassmannian manifold** on StyleGAN2-FFHQ. Extrinsic basis is a variant of Fréchet basis where the global semantic basis is discovered by the extrinsic mean of the Grassmannian manifold. In both scores, our Fréchet basis outperforms Extrinsic basis in 5 out 6 intermediate layers in the mapping network.

## F COMPARISON TO LOCAL BASIS

### F.1 QUANTITATIVE COMPARISON

In this section, we introduced a new experiment to quantitatively compare semantic factorization between Fréchet basis and Local Basis Choi et al. (2022b). Intuitively, this experiment measures the average of local DCI scores. To be more specific, consider $n$-samples $\{\mathbf{z}_i\}_{1 \leq i \leq n} \subset \mathcal{Z}$ of input Gaussian noise. (We set $n = 100$) Then, take $m$-samples from the neighborhood of each $\mathbf{z_i}$ and map them to the target latent space $\mathcal{W} = f(\mathcal{Z})$, where $f$ denotes the subnetwork from $\mathcal{Z}$ to $\mathcal{W}$:

$$\mathbf{w_{i,j}} = f(\mathbf{z}_i + \epsilon_{i,j}) \qquad \text{where} \quad \epsilon_{i,j} \sim \mathcal{N}(0, \sigma^2 I) \quad \text{and} \quad \mathbf{w}_i = f(\mathbf{z}_i), \tag{30}$$

for sufficiently small $\sigma > 0$. (We set $m = 1,000$ and $\sigma = 0.5$) Then, we measure DCI score for each neighborhood of $\mathbf{w}_i$, i.e., $\{\mathbf{w}_{i,j}\}_{1 \leq j \leq m}$, after representing these latent variables with each semantic basis as in Eq 13 (Fréchet basis and Local Basis at $\mathbf{w}_i$). The averages of these local DCI scores are compared between Fréchet basis and Local Basis. Below, we included the evaluated scores on $\mathcal{W}$-space of StyleGAN2, StyleGAN2-config-e, and StyleGAN1 trained on FFHQ. Note that the overall DCI scores are higher than Fig 4 because it is easier for a semantic basis to satisfy semantic consistency on the local region than on the entire latent space. As in the qualitative examples, Fréchet basis outperforms Local Basis in the quantitative assessment.

Table 2: **Quantitative Comparison of Semantic Factorization** by Local DCI ($\uparrow$). Local DCI score is evaluated at $\mathcal{W}$-space of each StyleGAN model.

| Model | Fréchet Basis | Local Basis |
|---|---|---|
| StyleGAN2 | **0.740** | 0.665 |
| StyleGAN2-e | **0.726** | 0.704 |
| StyleGAN1 | **0.733** | 0.677 |

### F.2 QUALITATIVE COMPARISON

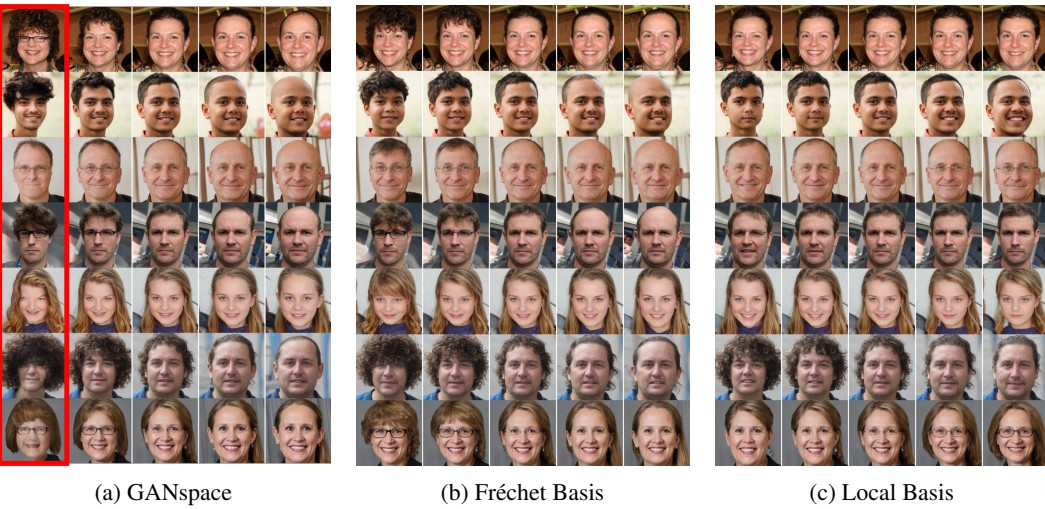

(a) GANspace      (b) Fréchet Basis      (c) Local Basis

Figure 13: **Comparison of Latent Traversals on StyleGAN2-FFHQ.** We used the annotated GANSpace on the semantics of "Bald". The corresponding Fréchet Basis and Local Basis are chosen by the cosine similarity. The traversal images along GANSpace are more deteriorated than the other two bases. The red box indicates where the image deterioration occurred.

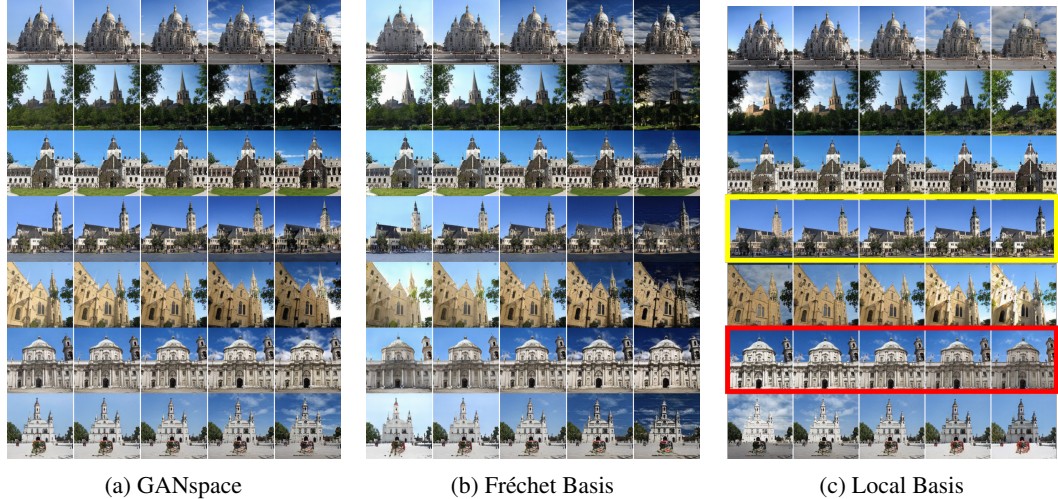

(a) GANspace        (b) Fréchet Basis        (c) Local Basis

Figure 14: **Comparison of Latent Traversals on StyleGAN2-LSUN Church.** We used the annotated GANSpace on the semantics of "Clouds". The corresponding Fréchet Basis and Local Basis are chosen by the cosine similarity. Some traversal examples along Local Basis do not show the annotated semantic variations compared to the other two bases. In the yellow box, no clouds appeared. In the red box, clouds appeared in all images.

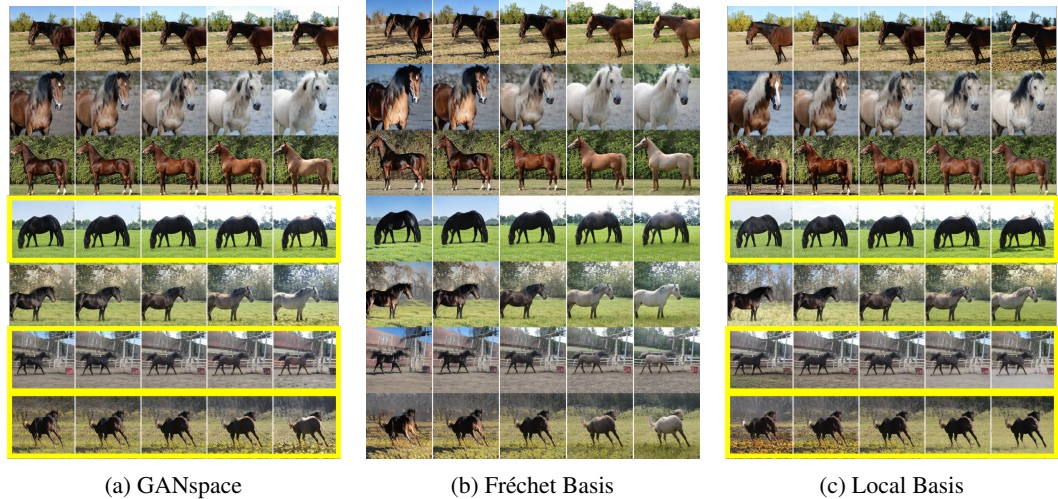

(a) GANspace        (b) Fréchet Basis        (c) Local Basis

Figure 15: **Comparison of Latent Traversals on StyleGAN2-LSUN Horse.** We used the annotated GANSpace on the semantics of "White horse". The traversal images of GANspace and Local Basis in the yellow box are less affected than that of Frechet Basis.

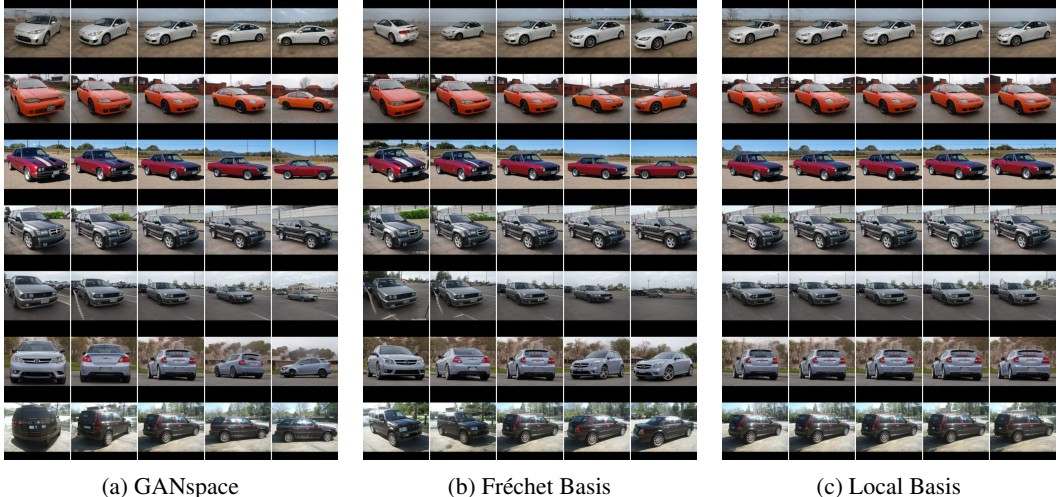

(a) GANspace       (b) Fréchet Basis       (c) Local Basis

Figure 16: **Comparison of Latent Traversals on StyleGAN2-LSUN Car.** We used the annotated GANSpace on the semantics of "Side to Front". The traversal images of Local Basis are not affected by perturbations.

# G    ADDITIONAL SEMANTIC FACTORIZATION COMPARISON

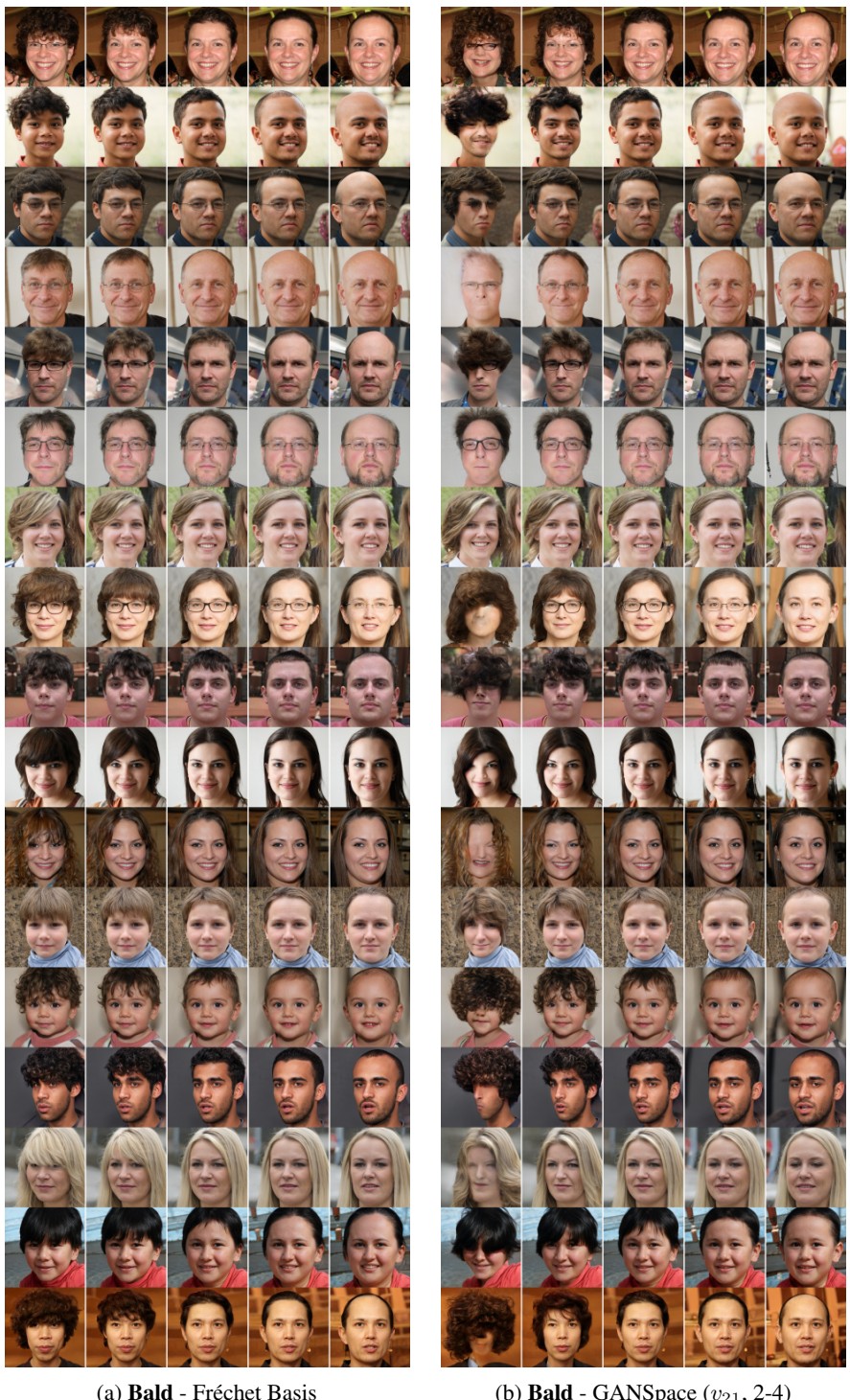

(a) **Bald** - Fréchet Basis                (b) **Bald** - GANSpace ($v_{21}$, 2-4)

Figure 17: **Comparison of Semantic Factorization on StyleGAN2-FFHQ** between Fréchet basis and GANSpace. ($v_i$, $l_1$-$l_2$) denotes the layer-wise edit along the $i$-th GANSpace component at the $l1$-$l2$ layers.

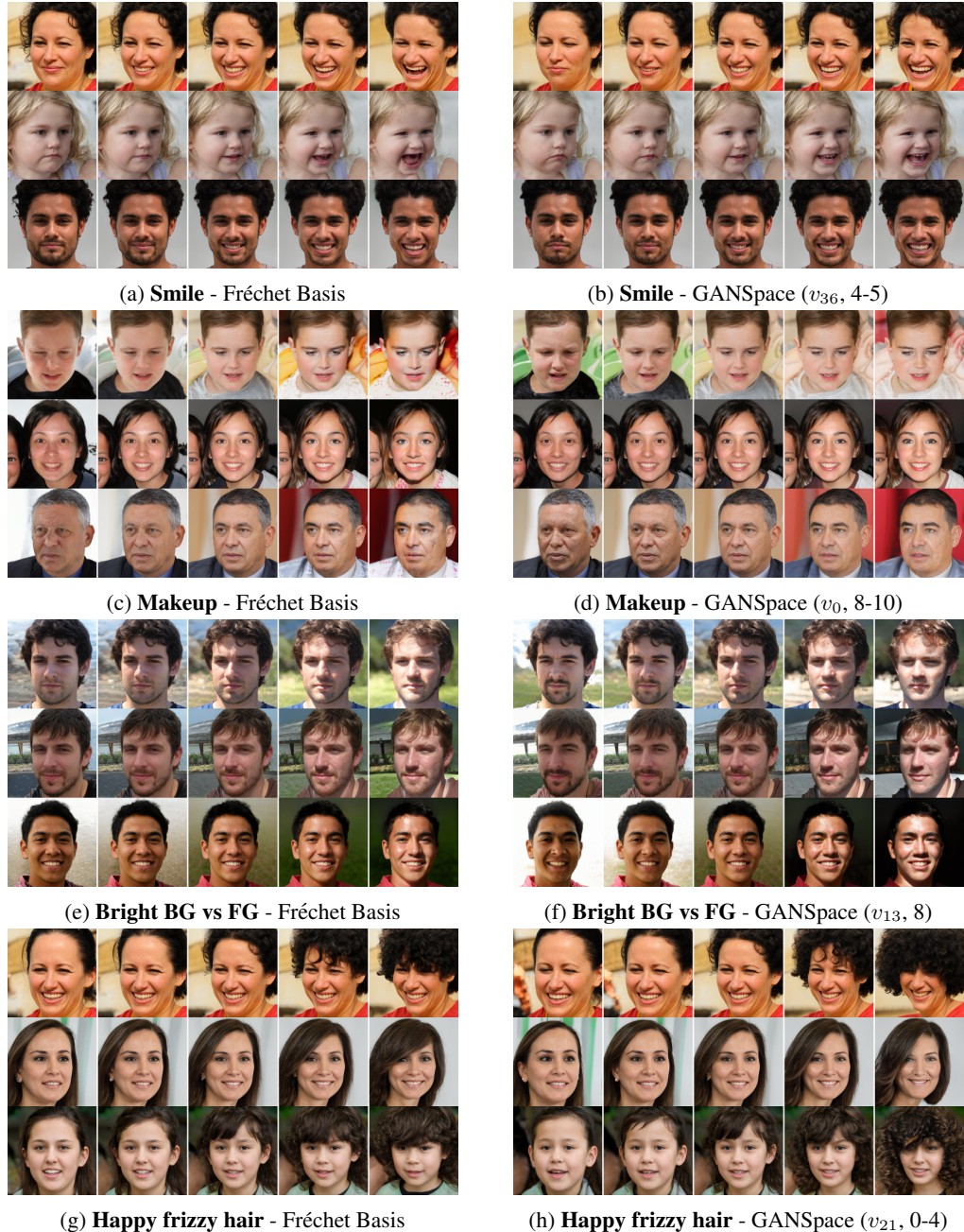

(a) **Smile** - Fréchet Basis

(b) **Smile** - GANSpace ($v_{36}$, 4-5)

(c) **Makeup** - Fréchet Basis

(d) **Makeup** - GANSpace ($v_0$, 8-10)

(e) **Bright BG vs FG** - Fréchet Basis

(f) **Bright BG vs FG** - GANSpace ($v_{13}$, 8)

(g) **Happy frizzy hair** - Fréchet Basis

(h) **Happy frizzy hair** - GANSpace ($v_{21}$, 0-4)

Figure 18: **Comparison of Semantic Factorization on StyleGAN2-FFHQ** between Fréchet basis and GANSpace. ($v_i$, $l_1$-$l_2$) denotes the layer-wise edit along the $i$-th GANSpace component at the $l1$-$l2$ layers.

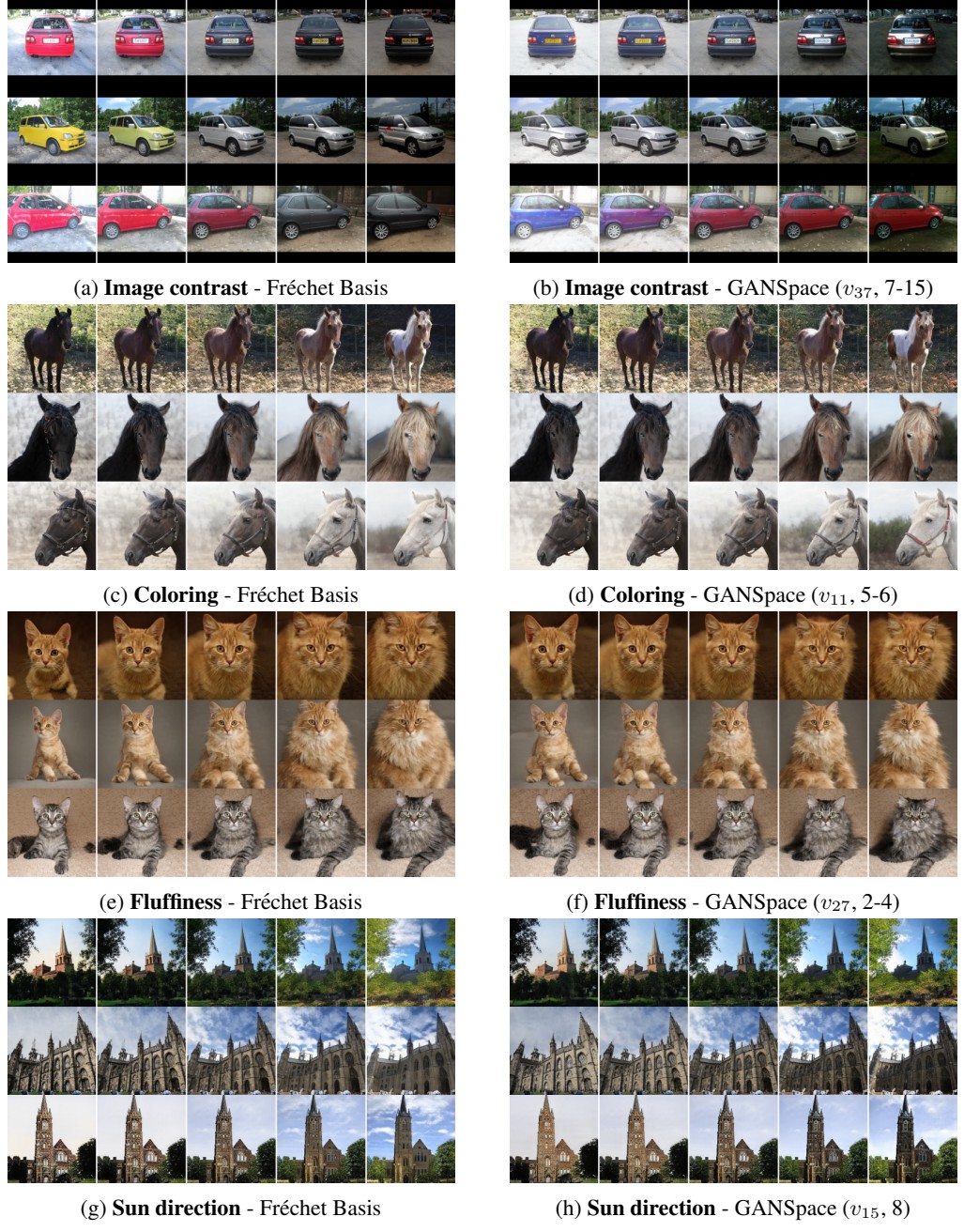

(a) **Image contrast** - Fréchet Basis

(b) **Image contrast** - GANSpace ($v_{37}$, 7-15)

(c) **Coloring** - Fréchet Basis

(d) **Coloring** - GANSpace ($v_{11}$, 5-6)

(e) **Fluffiness** - Fréchet Basis

(f) **Fluffiness** - GANSpace ($v_{27}$, 2-4)

(g) **Sun direction** - Fréchet Basis

(h) **Sun direction** - GANSpace ($v_{15}$, 8)

Figure 19: **Comparison of Semantic Factorization** between Fréchet basis and GANSpace. ($v_i$, $l_1$-$l_2$) denotes the layer-wise edit along the $i$-th GANSpace component at the $l1$-$l2$ layers. The image traversals are performed on StyleGAN2 (LSUN Car, Horse, Cat, and Church).

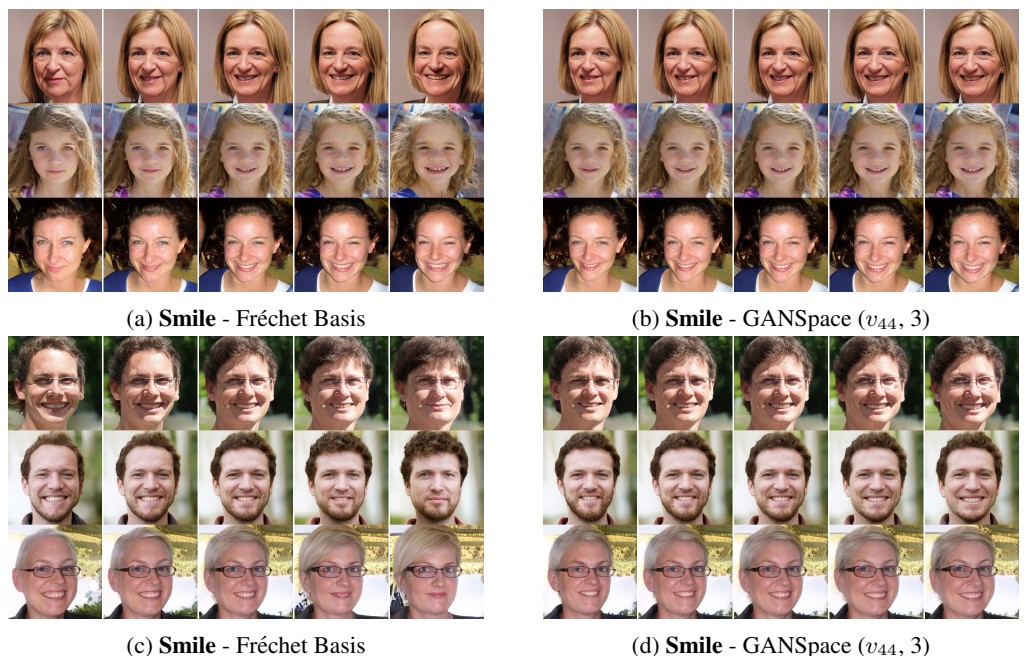

(a) **Smile** - Fréchet Basis

(b) **Smile** - GANSpace ($v_{44}$, 3)

(c) **Smile** - Fréchet Basis

(d) **Smile** - GANSpace ($v_{44}$, 3)

Figure 20: **Comparison of Semantic Factorization** between Fréchet basis and GANSpace. ($v_i$, $l_1$-$l_2$) denotes the layer-wise edit along the $i$-th GANSpace component at the $l1$-$l2$ layers. The image traversals are performed on StyleGAN1-FFHQ.

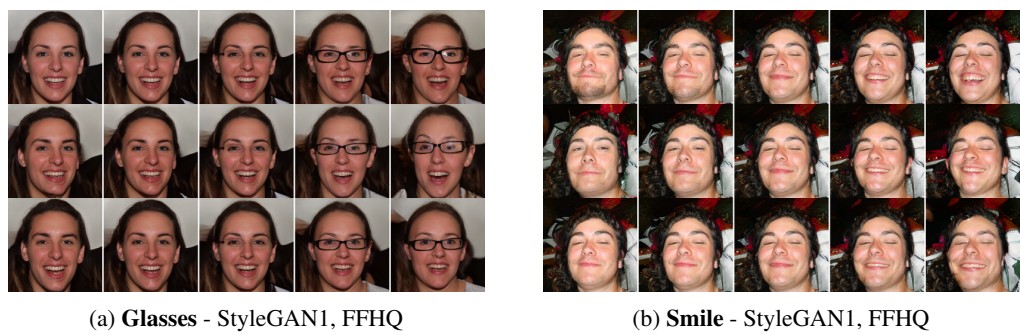

(a) **Glasses** - StyleGAN1, FFHQ

(b) **Smile** - StyleGAN1, FFHQ

Figure 21: **Comparison of Supervised method (InterfaceGAN) and Unsupervised methods (GANspace, Fréchet Basis).** The first row shows the results of supervised method (InterfaceGAN (Shen et al., 2020)), the second row shows GANspace results, and the last shows our traversing results along Fréchet Basis.

