# OpenReview forum: "Finding the Global Semantic Representation in GAN through Fréchet Mean"
_ICLR.cc/2023/Conference — ICLR 2023 poster_

### Official Review · Reviewer_G2MK · 2022-10-22

**Confidence:** 2
**Correctness:** 4
**Technical Novelty And Significance:** 3
**Empirical Novelty And Significance:** 2
**Recommendation:** 8

**Clarity, Quality, Novelty And Reproducibility:**

This seems to be a high quality paper. The authors did not clarify if they intend to release code and/or models. The authors improve upon previous work but present nice mathematical analysis backed up by good results.

**Details Of Ethics Concerns:**

no ethics concerns

**Strength And Weaknesses:**

## Strength
Results are very good visually compared to the GANSpace method which already gives very good results. The quantitative results are also superior to next best GANSpace. The paper is written in a clear and high quality fashion. The paper suggest nice mathematical insight on the GANs latent space decomposition and semantic structure. The mathematical formulations also are being backed up with the qualitative and qualitative results.

## Weaknesses
The paper is mathematically heavy where it is hard to follow (for me) on this translates the latent W (for example the 2D several layers and resolutions latent of StyleGAN2). I would have like to see more comparisons to a supervised method to understand the gap between the supervised and unsupervised methods.

**Summary Of The Paper:**

The authors suggest an unsupervised method for semantically meaningful perturbations of the learned latent W in GAN models such as StyleGAN. They propose to find a global basis called Frechet basis. The basis is discover in two steps: 1. The global semantic subspace is discovered by the Frechet mean in the Grassmannian manifold of the local semantic subspaces. 2.  Frechet basis is found by optimizing a basis of the semantic ´subspace via the Frechet mean in the Special Orthogonal Group. Additionally, the authors suggest a refinement scheme for previous methods.

**Summary Of The Review:**

I believe this paper is important to the community and advances the understanding of the semantic meaning of the GANs latent space.

---

> ### Author Response · Authors · 2022-11-14
> **Author response to Reviewer G2MK**
>
> **1. Generalization to the 2D latents**
>
> We appreciate the reviewer for the thoughtful comment. **Our framework can be applied to the 2D latent layer of StyleGANs by interpreting this layer as vectors of the same size, i.e., $\mathbb{R}^{r \times r} \rightarrow \mathbb{R}^{r^2}$. Moreover, we imagine that additional interesting properties might appear in this case.** That is, the spatial location of latent variables would correspond to that of corresponding semantics. For example, when a GAN model is trained on human facial images where eyes are aligned at a particular position, the latent variable at that position would represent the variation of eyes. In this case, the semantic basis for eyes can be discovered by analyzing a subset of latent variables, which is impossible for the fully connected latent layers. We consider this would also be an interesting future research.
>
> **2. Comparison to the supervised method**
>
> We compared Fréchet basis with InterfaceGAN [1]. InterfaceGAN is the supervised method for finding global semantic perturbations. In particular, the pre-trained attribute classifier from CelebA dataset is used to annotate generated images. The image traversal comparisons are presented in Fig 21 in the appendix F. For each semantic perturbation from InterfaceGAN, we compared two unsupervised methods (GANSpace, Fréchet basis) with the highest cosine similarity. While GANSpace shows image degradation, such as skewed glasses and distorted mouth, Fréchet basis achieves superior semantic factorization comparable to InterfaceGAN.
>
> > Added additional traversal results and highlighted in Olive at Fig 21 in Appendix F
>
> **References**
> [1] Yujun Shen, Jinjin Gu, Xiaoou Tang, and Bolei Zhou. Interpreting the latent space of gans for semantic face editing. In Proceedings of the IEEE/CVF Conference on Computer Vision and Pattern Recognition, pp. 9243–9252, 2020.

---

### Official Review · Reviewer_hbhj · 2022-10-24

**Confidence:** 4
**Correctness:** 4
**Technical Novelty And Significance:** 3
**Empirical Novelty And Significance:** 2
**Recommendation:** 6

**Clarity, Quality, Novelty And Reproducibility:**

- It is a bit unclear to me how global semantic directions as in the proposed method compare to local directions, and which one should be preferred in which situations. Explaining this could strengthen the motivation of the method.

- Figure 4, 5, 6: the X-axis represents discrete values, so I do not see why a line plot is used here. The connection between two consecutive X values is not meaningful.

**Strength And Weaknesses:**

** Strength **

- The paper presents a generalization of the intrinsic local tangents introduced by Choi et al. (2022a) by averaging the semantic subspaces. This is an interesting extension with solid theoretical support.


** Weakness **

1) The comparison against unsupervised global basis methods is not sufficient. Since this method is built upon Choi et al. 2022a, I think additional discussions and result comparisons with Choi et al. are needed despite it is a local method. By taking the mean of the local subspaces to discover global directions, does this lead to improvement or similar results?

2) There are not sufficient qualitative results in both the paper and the appendix. Figure 3 only shows the change of two attributes, and it is unclear how the method works for other attributes.

3) Results on additional datasets should be provided too, e.g., LSUN bedroom, cars, etc.

4) More analysis on the factorization of different attributes should be added. For example, consider common attributes such as pose, age, smile, eye gaze, wearing glass, etc., with the proposed unsupervised factorization, how these attributes are disentangled in the discovered global directions? Computing some correlations between these attributes might help.

4) Regarding robustness (Figure 5), it can be seen that Frechet means is not always better. Did the authors have any insight into when we can use GANspace and when to use Frechet means?

**Summary Of The Paper:**

The paper presents a method to disentangle the semantic space in a GAN model by using Frechet means. The pipeline has two steps. First, the semantic subspace is constructed by the Frechet mean in the Grassmannian manifold of the intrinsic tangent spaces. Then, the Frechet basis of the semantic subspace is constructed by using Frechet mean in the Special Orthogonal Group. The experimental results show that the proposed method produces better semantic factorization than the previous methods.

**Summary Of The Review:**

The paper presents an interesting and technically solid method to enable global semantic factorization of the latent space in GAN models, but the experiments are insufficient to justify the usefulness of the method. Given the current scale of the experiments and the quality of the results, the paper might need another cycle for major revision. I therefore lean toward a clear rejection.

---

> ### Author Response · Authors · 2022-11-14
> **Author response to Reviewer hbhj 2/2**
>
> **6. Graph Presentation**
> * Fig 4-5, 9-11: We thank the reviewer for the detailed advice. We agree with the reviewer that the scatter plot is a better graph to present our experimental results in Fig 4-5 and 9-11. We corrected Fig 4-5 and 9-11 accordingly.
> * Fig 6: The two consecutive $X$ values in Fig 6 are meaningful because it denotes the interpolation index from Fréchet basis ($i=0$) to GANSpace ($i=7$). Hence, we kept Fig 6 as a line plot.
>
> > Revised Fig 4-5, 9-11 and highlighted in Orange
>
> We hope that our response can address your main concerns. If so, we would like to kindly ask the reviewer to consider raising the score accordingly.
>
> **References**
> [1] Jaewoong Choi, Junho Lee, Changyeon Yoon, Jung Ho Park, Geonho Hwang, and Myungjoo Kang. Do not escape from the manifold: Discovering the local coordinates on the latent space of GANs. In ICLR, 2022.
> [2] Cian Eastwood and Christopher KI Williams. A framework for the quantitative evaluation of disentangled representations. In ICLR, 2018.

---

> ### Author Response · Authors · 2022-11-14
> **Author response to Reviewer hbhj 1/2**
>
> **1. Comparison to Local Basis [1]**
>
> We appreciate the reviewer for the valuable advice. **We compared Fréchet basis with Local Basis [1] in StyleGAN2 models trained on FFHQ, LSUN Church, LSUN Horse, and LSUN Car.** The corresponding components are selected using the cosine similarity to the annotated GANSpace. The results are presented in Fig $13-16$ in the appendix F. In our experiments, our Fréchet basis achieves better semantic factorization than Local Basis. This is interesting considering that Fréchet basis is an average of Local Basis. We guess this is because averaging the local information is beneficial for finding the global trend in the latent space.
>
> > Added additional traversal results and highlighted in Orange at Fig 13-16 in Appendix F
>
> **2. Additional Qualitative Examples**
>
> We appreciate the reviewer for the thoughtful comment. We included additional qualitative examples to the appendix F to strengthen the quantitative results provided in the paper.
> * **Additional Attribute** -  Smile, Makeup, Bright Background vs. Foreground, and Happy fizzy hair on StyleGAN2-FFHQ in Fig 18 of the appendix F.
> * **Additional Dataset** - LSUN Church (Fig 14, 19(g)), LSUN Horse (Fig 15, 19(c)), LSUN Car (Fig 16, 19(a)), and LSUN Cat (Fig 19 (e))
>
> > Added additional traversal results and highlighted in Orange at Fig 14-16, 18-19 in Appendix F
>
> **3. More analysis on the factorization of different attributes should be added.**
>
> **We believe the DCI score in the manuscript conducts a similar evaluation of the factorized semantics.** We think the suggested correlation analysis is for the quantitative comparison of semantic factorization. The DCI score is a supervised disentanglement metric that assesses the axis-wise alignment of semantics. To be more specific, the DCI score computes the importance of each axis for predicting the corresponding semantic variation. Then, the DCI score (Disentanglement score in [2]) is the average of the difference from one of the normalized importance entropy. Intuitively, the DCI score evaluates the degree to which each axis is concentrated on one semantics. We hope the presented DCI score can address the reviewer's concerns.
>
> **4. Regarding robustness (Figure 5), it can be seen that Fréchet means is not always better. Did the authors have any insight into when we can use GANspace and when to use Fréchet means?**
>
> We appreciate the reviewer for the insightful comment. **Compared to GANSpace, Fréchet basis is more robust for outliers in the latent space.** If the latent space is tightly clustered, GANSpace may achieve higher robustness than Fréchet basis. Specifically, GANSpace is the principal component of latent variables obtained by PCA. If there is an outlier latent variable that is far from the centroid, GANSpace is largely affected by this outlier. On the other hand, Fréchet basis is the Fréchet mean of the intrinsic tangent spaces. Hence, Fréchet basis is robust against the outlier in the latent variable position. If there is an outlier in terms of the intrinsic tangent spaces, it is a violation of the existence of global semantic perturbations.
>
> **5. Global semantic direction vs. Local semantic direction**
>
> **The semantic property of latent space determines whether the global method or local method is more appropriate.** If the global semantic perturbations exist on the target latent space, their global semantic consistency is appealing. It is enough to check one sample to know the semantic distribution of the entire latent space. However, it usually does not hold for the arbitrary latent space, even for the renowned $\mathcal{W}$-space in StyleGAN [1]. **Therefore, as described in the second paragraph of Sec 1, the proposed global methods are attempts to find the best-possible semantic basis on the target latent space.** There is a trade-off between the global methods and local methods. The local methods are more appropriate if the target latent space does not provide proper global semantic consistency. But we have to check each sample to discover the corresponding semantics. On the other hand, if the target latent space achieves reasonable global semantic consistency, the global methods would be a better option. Nevertheless, we believe that global methods should be investigated to induce and achieve a latent space with higher global semantic consistency.

---

### Official Review · Reviewer_McwQ · 2022-10-24

**Confidence:** 4
**Correctness:** 3
**Technical Novelty And Significance:** 3
**Empirical Novelty And Significance:** 3
**Recommendation:** 6

**Clarity, Quality, Novelty And Reproducibility:**

What is the n in eq. 1, and section 3.1.1? Is it d_W or the number of samples from Z? Do you consider the case where the number of samples of Z goes to infinity? What does the mu <= in eq. 11 stand for?

**Strength And Weaknesses:**

The paper shows that the Frechet basis can achieve better semantic factorization compared to stat-of-the-arts. The paper is well written, and the results are solid.
- I am just not sure on how the results in Section 4 change when one uses a different set of layers in the considered GAN, in particular how it depends on the size of each layers, and the nature of each layer (e.g. convolutional or fully connected). Could you provide more discussions in Section 4.1 or later?

**Summary Of The Paper:**

This paper proposes a way to find global semantic representation in GAN. It is based on looking for an average subspace among the tangent spaces in some layer of the generator. This subspace is then used to generate semantically factorized images in experiments, with both quantitative and qualitative results.

**Summary Of The Review:**

Rev: I would remain my score 6 as it would have been better to include the details of i.i.d. sampling in the evaluation of FID scores. As the problem has no uniqueness solution, I find that the impact of the results remains limited.

---

> ### Author Response · Authors · 2022-11-14
> **Author response to Reviewer McwQ**
>
> **1. Fréchet basis on various layers**
>
> We appreciate the reviewer for the constructive comment. In this paper, Fréchet basis is evaluated on the intermediate layers of the mapping network in pre-trained StyleGANs. Unfortunately, all intermediate layers are fully connected layers of the same size (Number of nodes = 512). Moreover, the previous works on the semantic perturbation in StyleGANs are focused on the fully connected layer or its concatenation, e.g., $\mathcal{W}$-space, $\mathcal{W}^{+}$-space, and $\mathcal{S}$-space.
> Therefore, we can only guess the behavior of Fréchet basis on the different layer structures. **First, if the size of each layer gets bigger, the local semantic subspaces may show a higher variance because they have a larger ambient space.** This makes it difficult for a latent space to achieve a proper degree of global semantic consistency. However, if the latent space achieves it, the larger ambient space would not be a problem because these local semantic subspaces are embedded in the lower dimensional submanifold. Moreover, the bigger latent space can provide a possibility of representing more diverse semantic variations. **Second, the CNN latent space is difficult to say because Fréchet basis is the average of the uninvestigated CNN's local semantic perturbations.** Motivated by the texture-bias of CNN features [1], we think Fréchet Basis on CNN would focus more on the lower-level features such as texture. Also, we imagine the spatial location of latent variables would correspond to that of corresponding semantics. For example, when a GAN model is trained on human facial images where eyes are aligned at a particular position, the latent variable at that position would represent the variation of eyes. In this case, the semantic basis for eyes can be discovered by analyzing a subset of latent variables, which is impossible for the fully connected latent layers.
>
> **2. Clarification of Symbols**
>
> Thank you for the detailed advice. We corrected $n$, which was used twice for different meanings, and added a description to $\mu \leq$.
> * $n \rightarrow m=\min(d_{\mathcal{Z}}, d_{\mathcal{W}})$ in Eq 1; This $m$ denotes the number of singular vectors for $df_{\mathbf{z}}$.
> * $n$ in Sec 3.1.1 (inline equation above Eq 7); This $n$ indicates the number of samples from $\mathcal{Z}$. In this paper, we did not consider how the estimated Fréchet basis changes as we increase the number of samples for calculating the Fréchet mean. Instead, we fixed the number of samples to 1,000 in all experiments, as described in Sec 4.1. We believe 1,000 samples are enough to estimate the Fréchet mean because a properly disentangled latent space has a small variance of the local semantic subspaces [2]. This leads to the small diameter of the sampled local intrinsic subspaces, which makes the Fréchet mean optimization for the global semantic subspace sample-efficient. Nevertheless, we agree with the reviewer that investigating the relationship between sample-efficiency and disentanglement would be an interesting future research.
> * $\mu \leq \mathbb{R}^{d_{\tilde{\mathcal{W}}}}$ in Eq 11; We used the symbol $\leq$ to denote that $\mu$ is a subspace of $\mathbb{R}^{d_{\tilde{\mathcal{W}}}}$. We agree with the reviewer that this notation requires specification. Hence, we added a description to Eq 11.
>
> > Highlighted the revisions in Blue
>
> We hope that our response can address your main concerns. If so, we would like to kindly ask the reviewer to consider raising the score accordingly.
>
> **References**
> [1] Robert Geirhos, Patricia Rubisch, Claudio Michaelis, Matthias Bethge, Felix A. Wichmann, and Wieland Brendel. ImageNet-trained CNNs are biased towards texture; increasing shape bias improves accuracy and robustness. CoRR, abs/1811.12231, 2018. 1, 4
> [2] Jaewoong Choi, Geonho Hwang, Hyunsoo Cho, and Myungjoo Kang. Analyzing the latent space of gan through local dimension estimation. arXiv preprint arXiv:2205.13182, 2022

---

> > ### Comment · Reviewer_McwQ · 2022-11-17
> > **1+ question**
> >
> > Thanks for your reply.
> >
> > As you use n samples of Z to estimate a global basis, I'd like to know in your evaluation, are you using an independent set of samples of Z to generate images and to compute FID scores etc, such as in Fig 3 and Table 1 ? This is not very clear to me by reading the paper, as for example you seem to use 3 fixed samples of Z to generate the middle images in each sub-figure of Fig 3. Are these 3 fixed samples in the n samples (to estimate a global basis)? If not, are you choosing them in some iid manner ?
> >
> > Also I'd like to know whether the global semantic subspace is unique or not, i.e. the solution of eq. (8) has a unique solution? Is it known in the literature? If the solution is not unique, I am concerned of the stability of the results as each time the interpretation would be quite different.

---

> > > ### Author Response · Authors · 2022-11-18
> > > **Thank you for the follow-up.**
> > >
> > > Thank you for the follow-up. **We used i.i.d. samples from $\mathcal{Z}$ to compute FID in Fig 5 and Table 1.** Then, the latent variables on $\mathcal{W}$ are obtained by feeding these samples to the subnetwork from $\mathcal{Z}$ to $\mathcal{W}$. Actually, this is how the distribution on $\mathcal{W}$ is defined. The image fidelity of each semantic basis is compared by measuring FID scores of image traversals from these i.i.d. samples. **Also, we employed this i.i.d. sampling for estimating Fréchet basis. The three latent variables in Fig 3 are chosen independently regardless of FID or Fréchet basis estimation.** We compared the semantic factorization of two semantic basis by traversing from the same starting images. We appreciate the detailed comment. These i.i.d. descriptions are added to the manuscript.
> > >
> > > From a theoretical perspective, the uniqueness of Fréchet mean is guaranteed on the Grassmannian manifold if the diameter of samples is less than $\pi/4$ [1]. However, this is a sufficient condition, not the equivalent condition. Unfortunately, the local semantic spaces from StyleGANs do not satisfy this sufficient condition in practice. In our additional experiments, the geodesic distance between two random local semantic spaces shows a mean of 7.14 and a standard deviation of 0.230 in $\mathcal{W}$-space of StyleGAN2-FFHQ. We also measured the variation in Fréchet mean resulting from running gradient descent on these local semantic spaces (Eq 8). We ran the optimization 5 times for 1000 local semantic subspaces, and assessed the geodesic distance between each pair of estimated Fréchet means. **The geodesic distance between these two Fréchet means shows an average of 0.555 and a standard deviation of 0.186.** In this regard, we believe that the Fréchet mean optimization for the global semantic subspace can provide a stable estimate.
> > >
> > > **References**
> > > [1] Evgeni Begelfor and Michael Werman. Affine invariance revisited. In 2006 IEEE Computer Society Conference on Computer Vision and Pattern Recognition (CVPR’06), volume 2, pp. 2087–2094. IEEE, 2006.

---

### Official Review · Reviewer_toHN · 2022-10-25

**Confidence:** 4
**Correctness:** 3
**Technical Novelty And Significance:** 2
**Empirical Novelty And Significance:** Not applicable
**Recommendation:** 8

**Clarity, Quality, Novelty And Reproducibility:**

### Clarity
In general, the paper is well-written with sufficient technical details.
### Novelty
The proposed method seems to have incremental novelty.
### Reproducibility
Codes are provided in the supplementary materials. However, I'm unsure whether it is sufficient to reproduce all the experiments in the paper based on current information.

**Strength And Weaknesses:**

### Strength
* The paper is well-written and easy to follow. Moreover, the paper provides a thorough background introduction and related prior works.
* The motivation is clear, and the intuition behind the proposed method is straightforward. Interpreting the global semantic variation as the mean of the local semantic variations seems simple but effective.
* The experiment results seem promising. The proposed method appears effective and provides better semantic factorization and robustness.

### Weaknesses
* I have some concerns about the design choices in the proposed method. I feel more analysis should be conducted. For example, there are plenty of ways to find the mean of subspaces; why would Frechet mean be the best choice? It would be better to provide an ablation study on such a design.
* The refinement improvement on GANSpace seems limited. While the refinement on SeFa leads to a near Frechet basis performance. Is there any justification or explanation for the such result?

**Summary Of The Paper:**

This paper proposes an unsupervised way to find the global semantic perturbations in a latent space in GAN, named Frechet Basis. Specifically, it is found by using Frechet mean to the local semantic perturbations. Experiments show that the proposed basis can provide better semantics and robustness than the previous unsupervised global methods. The paper also introduces a basis refinement scheme that can refine prior works by running the basis refinement on their subspaces.

**Summary Of The Review:**

In general, the paper proposes a straightforward solution for an important problem. However, the experiments seem limited. I am willing to raise my recommendation if more theoretical justification or empirical study are provided.

---

> ### Author Response · Authors · 2022-11-14
> **Author response to Reviewer toHN**
>
> **1. Why is Fréchet mean the best choice for finding the mean of subspaces?**
>
> We appreciate the reviewer for the thoughtful advice. Following the reviewer's advice, **we tested an alternative to finding the mean of subspaces and added the results at Fig 12 in the appendix E.** Specifically, we tried the extrinsic mean $\mu_{E}$ on Grassmannian manifold [1]. (The definition of the extrinsic mean is provided in the next paragraph.) We refer to this alternative as the *Extrinsic basis*.
> The intrinsic Fréchet mean $\mu_{fr}$ and the extrinsic mean $\mu_{E}$ are two popular choices for defining the mean of subspaces. Before presenting the experimental results, we think it is worth noting that **the extrinsic mean depends on the embedding function $\Phi$.** Hence, we believe it is more natural to choose the Fréchet mean because it is defined solely by the intrinsic property of Grassmannian manifold. Moreover, the experiments support our claim. We compared the proposed Fréchet basis with Extrinsic basis on StyleGAN2 trained on FFHQ. FID and DCI scores of two basis are compared for each intermediate layer in the mapping network. The experimental results show that our Fréchet basis outperforms Extrinsic basis for 5 out of 6 layers in both scores.
>
> - **Definition of extrinsic mean** The extrinsic mean $\mu_{E}$ is defined as the minimizer of squared extrinsic metrics $d_{E}$, i.e., for $x_1, \ldots, x_{n} \in Gr(k, \mathbb{R}^{n})$,
> $$
>     \mu_{E} = \arg\min_{\mu \in X} \sum_{1\leq i\leq n} d_{E}
>     \left( \mu, x_{i}  \right)^{2},
>     \quad \textrm{ where } d_{E} \left( \mu, x_{i}  \right)
>     = d_{\Phi} \left( \Phi(\mu), \Phi(x_{i})  \right),
> $$
> where $\Phi$ denotes an appropriate embedding of $Gr(k, \mathbb{R}^{n})$. Following [2], we set the embedding $\Phi$ to be the corresponding projection $\mathbb{P}\_{x_{i}}$, i.e., $\Phi(x\_{i}) = \mathbb{P}\_{x\_{i}} = M\_{x\_{i}}^{\top} M\_{x\_{i}}$ where $M\_{x\_{i}} \in \mathbb{R}^{n \times k}$ indicates the column-wise concatenation of an orthonormal basis of $x_{i}$. Also, the Frobenius norm is adopted for $d\_{\Phi}$.
>
> > Added additional experiments and highlighted in Red at Fig 12 in Appendix E
>
> **2. The refinement improvement on GANSpace seems limited. While the refinement on SeFa leads to a near Fréchet basis performance. Is there any justification or explanation for the such result?**
>
> We appreciate the reviewer for the insightful comment. **The limited improvement on GANSpace is because the GANSpace and Fréchet basis share similar approaches.** Both methods analyze the learned latent manifold $\mathcal{W}$. To be more specific, [3] proved that Local Basis is equivalent to the Local PCA. Hence, Fréchet basis is intuitively the average of these Local PCA, and GANSpace is the global PCA. Although the semantic subspace of GANSpace is suboptimal, GANSpace discovers a near-optimal semantic basis in the subspace through this similarity. Therefore, the improvement from basis refinement is marginal for GANSpace.
> **On the other hand, SeFa takes a completely opposite approach to Fréchet basis.** SeFa investigates the first parameters applied to the target latent space $\tilde{\mathcal{W}}$, ignoring the learned latent manifold $\mathcal{W}\subset\tilde{\mathcal{W}}$. Therefore, the gain from basis refinement is more significant for SeFa. Interestingly, the refined SeFa achieved a similar-or-better score than the refined GANSpace. **We interpret this result as a promising sign for combining these two approaches.** As described above, these approaches are complementary to each other. Combining these two approaches, e.g., applying SeFa on the locally estimated $\mathcal{W}$, would be an interesting future research.
>
> **3. More detailed implementation detail**
>
> To improve the reproducibility of manuscript, we added a Section summarizing the implementation details in the appendix B.
>
> > Added Implementation detail and highlighted in Red in Appendix B
>
> We hope our replies addressed all the concerns. If our replies feel satisfactory, we would like to kindly ask the reviewer to consider raising the score.
>
> **References**
> [1] Anuj Srivastava and Eric Klassen. Monte carlo extrinsic estimators of manifold-valued parameters. IEEE Transactions on Signal Processing, 50(2):299–308, 2002.
> [2] Tim Marrinan, J Ross Beveridge, Bruce Draper, Michael Kirby, and Chris Peterson. Finding the subspace mean or median to fit your need. In Proceedings of the IEEE Conference on Computer Vision and Pattern Recognition, pp. 1082–1089, 2014.
> [3] Jaewoong Choi, Junho Lee, Changyeon Yoon, Jung Ho Park, Geonho Hwang, and Myungjoo Kang. Do not escape from the manifold: Discovering the local coordinates on the latent space of GANs. In International Conference on Learning Representations, 2022.

---

> > ### Author Response · Authors · 2022-12-08
> > **Thank you for the thoughtful feedback.**
> >
> > Thank you for the thoughtful feedback.
> >
> > We hope our addtiional ablation study on the mean of subspaces (1. in Response) was helpful in addressing the reviewer's concerns.
> >
> > If our responses were helpful, we would like to gently ask the reviewer to consider re-evaluating our manuscript.

---

> > > ### Comment · Reviewer_toHN · 2022-12-08
> > > **Response to Authors**
> > >
> > > Thank you for the response. My concerns are all well-addressed, and I am happy to raise my score.

---

> > > > ### Author Response · Authors · 2022-12-08
> > > > **Thank you.**
> > > >
> > > > Thank you for your prompt response.
> > > > We are glad our responses were helpful in addressing the reviewer's concerns. The reviewer's insightful questions led to significant improvement of our manuscript. Thank you again.

---

### Author Response · Authors · 2022-11-14
**Author response to all reviewers**

We deeply thank the reviewers for spending time reading our manuscript carefully and providing thoughtful feedback. We think that the reviewers raised several valuable questions, and answering those questions has significantly improved our work. Below we address specific questions and comments to each reviewer. We highlighted the corresponding revisions in the manuscript in Red for Reviewer toHN, Blue for Reviewer McwQ, Orange for Reviewer hbhj, and Olive for Reviewer G2MK.

---

### Decision · Program_Chairs · 2023-01-20

**Decision:**

Accept: poster

**Justification For Why Not Higher Score:**

Most of  the empirical evaluation focused on StyleGAN and StyleGAN2 models, with design choices on particular layers. It is not entirely obvious whether these results would also hold for other types of GANs. Moreover, the improvement over GANspace seems a bit more modest than at first glance.

**Justification For Why Not Lower Score:**

The theoretical contributions are quite substantial, and the empirical results presented are quite promising. That, and the reviewers' support of the paper are sufficient for the paper to be accepted.

**Metareview: Summary, Strengths And Weaknesses:**

The authors propose a new unsupervised method for find global semantic perturbations in a GAN latent space, called the Frechet basis. The core of the method is well motivated, the experimental results are promising, and the paper is clear and well written. The reviewers all agree the paper should be accepted, but a couple had caveats about somewhat limited novelty relative to previous works, and robustness to the method. While these constructive criticisms are fair, on the balance I think the paper is strong mainly because the strengths of the paper outweigh the weaknesses. The authors also improved the paper based on reviewer feedback, and as a result, the paper will be well received by an ICLR audience.

**Note From Pc:**

if the above contains the word "oral" or "spotlight" please see: "oral" presentation means -> notable-top-5% and "spotlight" means -> notable-top-25%. As stated in our emails, we are disassociating presentation type from AC recommendations